# Hippocampal neurons code individual episodic memories in humans

Luca D. Kolibius [1,2,3] ✉, Frederic Roux[3], George Parish [3], Marije Ter Wal [3], Mircea Van Der Plas [2,3], Ramesh Chelvarajah[3,4], Vijay Sawlani[4], David T. Rollings[4], Johannes D. Lang[5], Stephanie Gollwitzer[5], Katrin Walther [5], Rüdiger Hopfengärtner[5], Gernot Kreiselmeyer[5], Hajo Hamer[5], Bernhard P. Staresina [6,7], Maria Wimber [2,3], Howard Bowman[3,8] & Simon Hanslmayr [2,3] ✉

The hippocampus is an essential hub for episodic memory processing. However, how human hippocampal single neurons code multi-element associations remains unknown. In particular, it is debated whether each hippocampal neuron represents an invariant element within an episode or whether single neurons bind together all the elements of a discrete episodic memory. Here we provide evidence for the latter hypothesis. Using single-neuron recordings from a total of 30 participants, we show that individual neurons, which we term episode-specific neurons, code discrete episodic memories using either a rate code or a temporal firing code. These neurons were observed exclusively in the hippocampus. Importantly, these episode-specific neurons do not reflect the coding of a particular element in the episode (that is, concept or time). Instead, they code for the conjunction of the different elements that make up the episode.

Episodic memory refers to our ability to reinstate the what, where and when of past experiences[1]. This ability is thought to depend on the reinstatement of neural activity that was present at memory encoding[2]. It is undisputed that the hippocampus plays an integral role in episodic memory processing[3–5] and the binding of multimodal information[6]. However, how it codes episodic memories remains controversial.

One important open question is whether neurons in the hippocampus code for specific elements or an entire episode. Concept neurons in the hippocampus fire in response to specific invariant elements independent of the context in which they are presented[7–10]. One contemporary idea is that the diverse elements that make up an episode are coded by the simultaneous activity of a set of these concept neurons[11,12] or by expanding the selectivity of existing concept neurons[13]. According to this framework, when you are sitting in your favourite coffee shop with your best friend, one set of concept neurons might code for the coffee shop and a separate set for your friend (Fig. 1a).

Alternatively, single units in the hippocampus might sparsely encode a specific set of elements within an individual episode and act as pointers to cortical modules during memory reinstatement. According to this so-called indexing theory[14,15], the entire episode with your friend in the coffee shop is represented by a set of hippocampal neurons (Fig. 1a). Unlike concept neurons, these episode-specific neurons (ESNs) would fire in response to the conjunction of all the diverse information within an episode and not in response to individual content elements. Despite computational models pointing towards the

[1]Department of Biomedical Engineering, Columbia University, New York, NY, USA. [2]Centre for Cognitive Neuroimaging, School of Psychology and Neuroscience, University of Glasgow, Glasgow, UK. [3]Centre for Human Brain Health, School of Psychology, University of Birmingham, Birmingham, UK. [4]Complex Epilepsy and Surgery Service, Neurosciences Centre, Queen Elizabeth Hospital Birmingham, Birmingham, UK. [5]Epilepsy Center, Department of Neurology, University Hospital Erlangen, Friedrich-Alexander-Universität Erlangen-Nürnberg, Erlangen, Germany. [6]Department of Experimental Psychology, University of Oxford, Oxford, UK. [7]Oxford Centre for Human Brain Activity, Wellcome Centre for Integrative Neuroimaging, Department of Psychiatry, University of Oxford, Oxford, UK. [8]Centre for Cognitive Neuroscience and Cognitive Systems and the School of Computing, University of Kent, Canterbury, UK. ✉e-mail: luca.kolibius@columbia.edu; simon.hanslmayr@glasgow.ac.uk

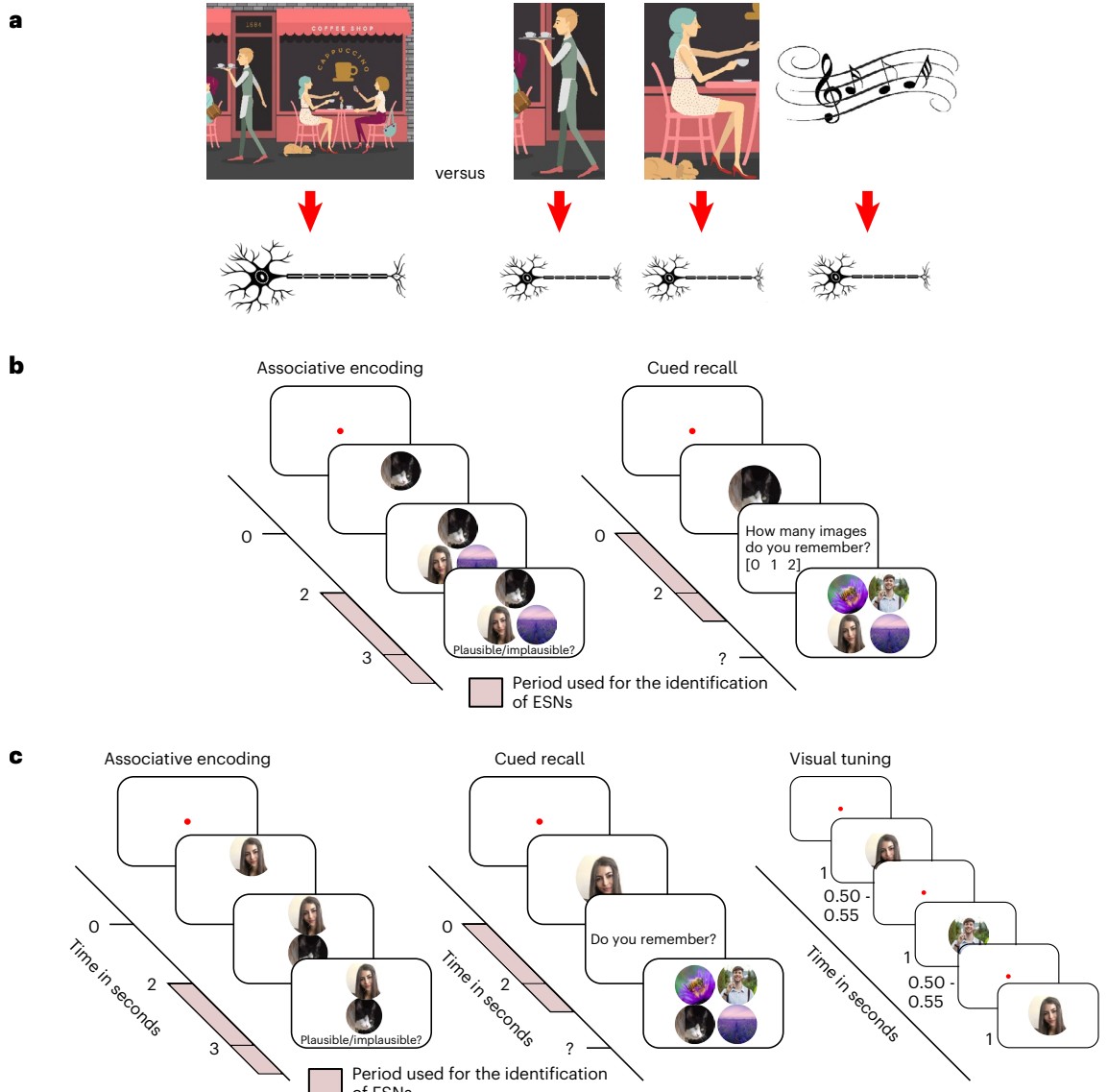

**Fig. 1 | Difference between indexing theory and concept-neuron-based hippocampal coding of episodic memories, and procedures for Experiments 1 and 2. a**, Left, the indexing theory proposes that hippocampal neurons represent a conjunctive code, acting as pointers to various elements of an episode (the smell of the coffee, your friend, the background music, the café and so on) rather than directly coding for the elements themselves[14]. Right, concept neurons in the hippocampus are thought to code for specific elements or concepts[7-9]. Within this framework, a group of neurons collectively code an episodic memory, with each neuron representing a specific element involved in that episode (that is, a neuron coding for the coffee, another neuron coding for your friend and so on[11,12]). It is important to note that one index or one concept is likely to be coded by an assembly of neurons, not a single neuron. **b**, Procedure for Experiment 1. The participants imagined a vivid episode with an animal cue and two associate images (two faces, two places or a face and

a place), rated its plausibility, and later recalled the associated images when cued with the animal cue. The experiment was self-paced, and each episode was learned and tested once. This approach is suitable for investigating episodic memory as originally defined by Tulving in 1972[29]. Following each encoding block of roughly 20 episodes, the participants performed a short distractor task. The pink areas represent the time windows used for subsequent analyses (Methods). **c**, Procedure for Experiment 2. Left, the memory task was largely the same as in Experiment 1 (**b**), but events had one cue and one associate image (both an animal, a face or a place). Right, after the memory task, the participants performed a memory-free visual tuning task where previously shown stimuli were presented multiple times in quick succession. This approach has traditionally been used to identify putative concept neurons. The original images have been replaced due to copyright issues.

existence of ESNs[16-19], to this day there is no evidence for such a sparse conjunctive code in humans.

In the present work, we provide support for the existence of this content-agnostic episodic memory code implemented through ESNs. We leveraged intracranial microwire recordings to investigate the firing patterns of neurons in the human hippocampus and hypothesized that a significant number of hippocampal neurons would reinstate their firing rate within a specific episode (that is, fire during encoding and retrieval).

Importantly, these ESNs would code for the conjunctive elements present within an episode and would not be tuned to individual elements within the episode. The existence of ESNs does not preclude concept neurons from participating in episodic memory processing. However, investigating the role of concept neurons in episodic memories goes beyond the scope of this work. As control analyses, we investigated whether this firing activity can be explained by a firing response to specific invariant elements, as occurs in concept neurons[9], or by a time preference, as occurs in time cells (TCs)[20,21].

## Results

We analysed recordings from two separate experiments (Experiment 1: 585 neurons in the hippocampus, 16 participants, 7 female, mean age 36.125 years, from 26 to 53 years; Experiment 2: 216 neurons in the hippocampus, 14 participants, 7 female, mean age 33.857 years, from 19 to 58 years) where the participants were implanted with stereotactic Behnke–Fried depth electrodes in the hippocampus (Supplementary Fig. 1), while they performed a memory association task (Fig. 1b,c).

During the encoding phase of Experiment 1, the participants created a vivid mental story consisting of an animal cue and two associate images (two faces, two places or a face and a place). By contrast, Experiment 2 consisted of one cue and one associate image (both an animal, face or place). The encoding and recall phases of the experiment were interleaved with a short distractor task where the participants had to judge whether a series of 15 numbers were odd or even. The distractor phase lasted between 22.43 s and 224.52 s (median duration, 42.19 s). During the recall phase, the animal cue was presented again, and the participants were asked to retrieve the associate image(s). The experiments were self-paced, and every episode was learned and retrieved only once. The participants correctly recalled on average 68.38% (s.e. = 4.64%) of episodes in the first experiment (Supplementary Table 1) and on average 65.63% (s.e. = 4.45%) of episodes in the second experiment (Supplementary Table 2). This is substantially more than would be expected by chance (16.7% and 25%, respectively).

### Identifying ESNs

For every neuron, we determined the firing rate during each episode at encoding and retrieval. We then $z$-scored the firing rate across all encoding and retrieval episodes and excluded all later forgotten episodes. This was done independently for encoding and retrieval to account for general differences in firing rates. We measured episode-specific firing reinstatement as the product of the standardized firing rates at encoding and retrieval (Fig. 2a).

Using an episode-shuffling procedure, we generated a distribution of reinstatement values expected by chance. A neuron was considered an ESN if (1) the empirical reinstatement value exceeded the 99th percentile of the shuffled distribution for at least one episode and (2) the standardized firing rate for encoding and retrieval of that episode each exceeded 1.645 ($P_{\text{right-tailed}} < 0.05$). The second criterion prevented the identification of ESNs that would excessively fire at only one phase of the task (that is, encoding or retrieval).

It could be argued that ESNs identified in this manner could reflect the firing of cells tuned to the image of the animal cue rather than the conjunction of all elements, since the cue is episode-unique and presented during encoding and retrieval. To address this issue, in Experiment 1, we excluded neurons that showed a significant firing increase during the first second after the encoding of the animal cue for episodes that were later reinstated (Methods). This procedure has traditionally been used to identify putative concept neurons[7–9]. Using this approach, we identified a significant number of hippocampal ESNs in Experiment 1 (136 out of 585 neurons (23.25%); $P < 0.001$; permutation test; Fig. 2b). Comparable results are obtained when (1) adding up the standardized firing rate between encoding and retrieval instead of multiplying them (E and R ≥ 1.645; reinstatement, E + R) (125 ESNs; $P < 0.001$), (2) increasing the minimum standardized firing rate from $z = 1.645$ to $z = 2.6$ (E and R ≥ 2.6; reinstatement, E × R) (29 ESNs; $P < 0.001$) and (3) using a different reinstatement measure that normalizes the encoding-and-retrieval product by their absolute difference (E and R ≥ 1.645; reinstatement, (E × R)/|E − R|) (53 ESNs; $P < 0.001$). This reinstatement measure has the important advantage of considering the similarity between the encoding and retrieval firing rates.

In Experiment 1, 117 of 136 ESNs (86.03%) coded for a single episode, whereas the rest coded for multiple episodes. Two example ESNs are shown in Fig. 3. These ESNs are unlikely to be concept cells tuned to the animal cue as the firing rate during encoding reaches its maximum only after the presentation of the associate stimulus (Fig. 4).

Note that the proportion of neurons that can be classified as ESNs is proportional to the number of events learned and retrieved (the same is the case for concept neurons). This is because we apply the threshold derived from the first permutation test to all episodes, without family-wise error correction. It is therefore not suitable for determining the sparseness of the hippocampal code. However, the proportion of ESNs of all recorded neurons is useful as an estimation of how many ESNs we can expect in future analyses.

It is crucial to understand that this $\alpha$-level inflation does not extend to the group-level permutation test, where the same number of tests are applied to randomly shuffled data. We have added a simulation using random values as spike rates and using circularly shuffled spike times to show that there is no inflation of the $\alpha$ error at the group level at which we interpret our findings (Methods and Supplementary Fig. 2).

ESNs are suggested to reflect a unique coding mechanism of the hippocampus[14,15]. In line with this, we did not find a significant number of ESNs in the parahippocampus (15 out of 104 neurons; $P = 0.5396$; permutation test). However, it should be noted that all 104 parahippocampal neurons originate from only five different microwire bundles over 14 sessions in five different participants, and therefore these results (albeit in line with the indexing theory) should be interpreted with caution. The second experiment did not contain a significant number of ESNs in the parahippocampus (3 out of 25 neurons; $P = 0.1199$).

To conclude, we found a significant number of ESNs in the hippocampus but not in the parahippocampus. The analysis approach we used to identify ESNs is robust to deviations in the parameter space.

### ESNs do not code for the content/visual properties of images

Traditionally, visually responsive neurons have been identified using the repeated presentation of a stimulus. In the above analysis, we presented the animal cue only once, which is suboptimal for ruling out concept neurons tuned to the animal cue. To ameliorate this shortcoming, in Experiment 2 we added a visual tuning task (Fig. 1c) after the memory association task. During the visual tuning task, images from the memory task were repeatedly shown in quick succession. This approach is widely used to identify putative concept neurons that respond to one of the images independently of any memory processes (for example concept neurons, see Supplementary Fig. 3)[7–9]. When excluding concept neuron activity in this independent dataset, we replicated our previous results and identified a significant number of ESNs (38 out of 216 neurons (17.59%); $P = 0.0053$; permutation test; Fig. 2c). In Experiment 2, 34 of 38 ESNs (89.47%) coded for a single episode, whereas the rest coded for multiple episodes.

However, traditional concept neuron detection methods might be too conservative to identify weakly tuned concept neurons. To address this concern, we drastically reduced the threshold of what constitutes a concept neuron—that is, lowering the uncorrected threshold from $P = 0.0005$ to $P = 0.05$—which increased the number of concept neurons from 58 to 155 (out of 216 neurons). During a typical tuning task, an average of 108.7 (minimum, 80; maximum, 156) different images are shown, and each image is tested for visual tuning. There is no correction for multiple testing, rendering a threshold at $P < 0.05$ very liberal.

Remarkably, incorporating this liberal threshold to exclude potential concept neurons had little effect on the number of ESNs, which remained almost unchanged (36 out of 216 neurons (16.67%); $P = 0.0025$; permutation test). It is conceivable that some images that are presented during the visual tuning task act as cues that reactivate some ESNs. These reactivated ESNs would then be erroneously rejected as concept neurons. However, in practice, only four potential ESNs

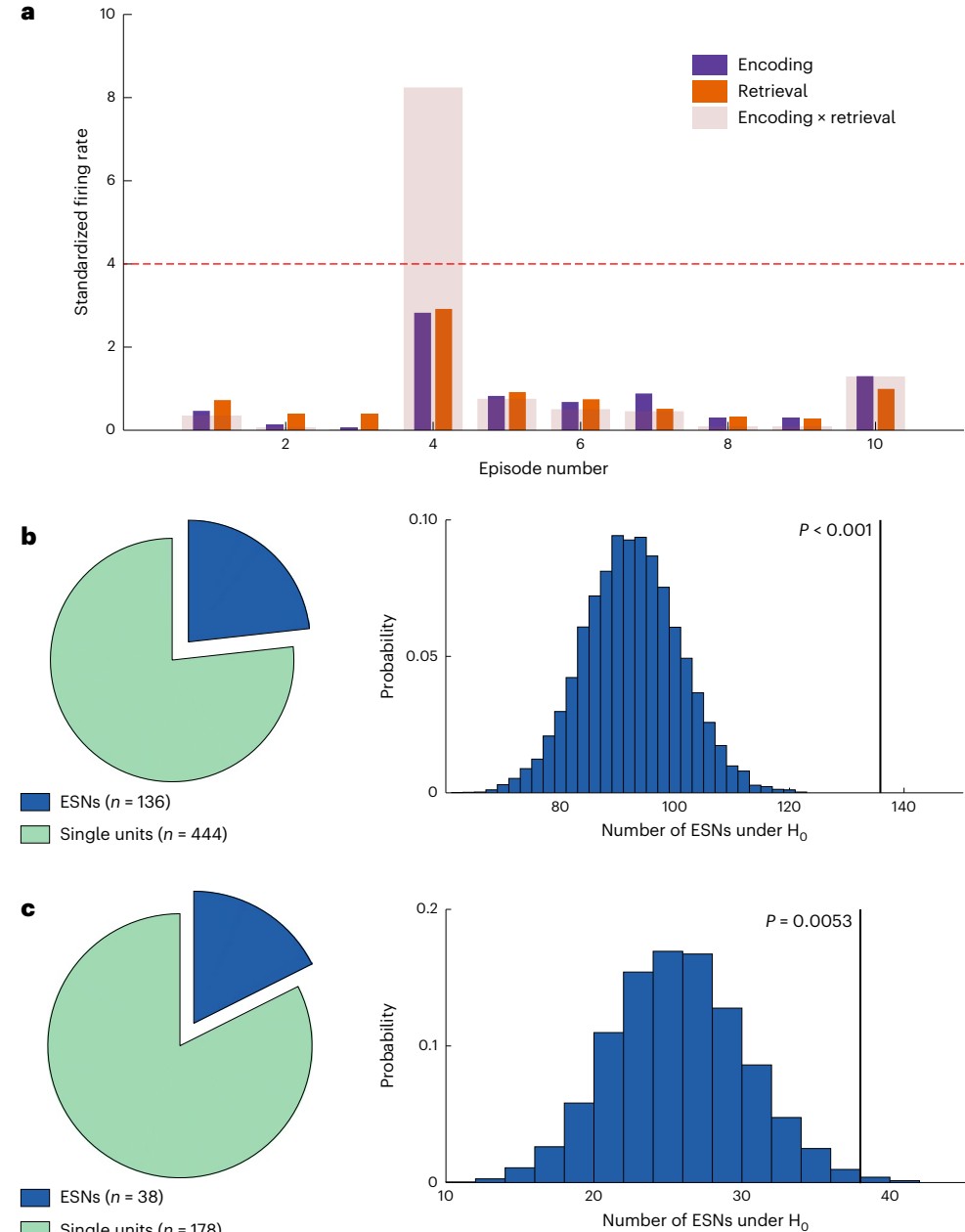

**Fig. 2 | Analysis schematic and number of ESNs identified in Experiments 1 and 2. a**, A schematic for identifying ESNs. The plot shows the *z*-scored firing rate on the *y* axis for ten simulated episodes on the *x* axis colour-coded for encoding and retrieval (purple and orange, respectively). The transparent bars encompassing encoding and retrieval indicate the product of encoding and retrieval firing rates, which is used as the measure of episode-specific firing reinstatement. The dashed red line shows the threshold (derived from a shuffling procedure; Methods). Because of the way ESNs are defined, they are required to fire substantially above their average firing rate during encoding and retrieval, which rules out neurons that generally show an increased firing rate during remembered episodes. **b**, Identified ESNs during Experiment 1. Left, the number of ESNs that showed significant firing reinstatement to at least one episode (dark blue) and the number of neurons that showed no firing reinstatement (green). Right, the number of ESNs expected by chance and the empirical number of ESNs (136 out of 585 neurons; $P < 0.001$; one-sided permutation test). **c**, Same as **b** but for Experiment 2. Of a total of 216 hippocampal neurons, we identified 38 ESNs ($P = 0.0053$; one-sided permutation test).

were excluded on the basis of the visual tuning task (six when lowering the concept neuron threshold). We suspect that ESNs were not reactivated during the visual tuning task because the participants were not instructed to actively retrieve memories and thus may not have been in a "retrieval mode"[22,23]. Tulving first proposed this concept in 1983[24,25], referring to the cognitive state that occurs when we actively attempt to remember something. Being in a retrieval mode increases the likelihood that a memory cue successfully triggers memory retrieval. Nonetheless, we cannot rule out that in some cases ESNs were reactivated

during the visual tuning task and subsequently rejected. However, this would only make our analysis for identifying ESNs more conservative.

## ESNs are limited to later remembered episodes

We have so far demonstrated that ESNs reinstate their firing rate when remembering a unique episode. This reinstatement cannot be explained by the semantic content or visual properties of the used image, which strengthens the notion that ESNs code for memories. In line with this, we did not find a significant number of

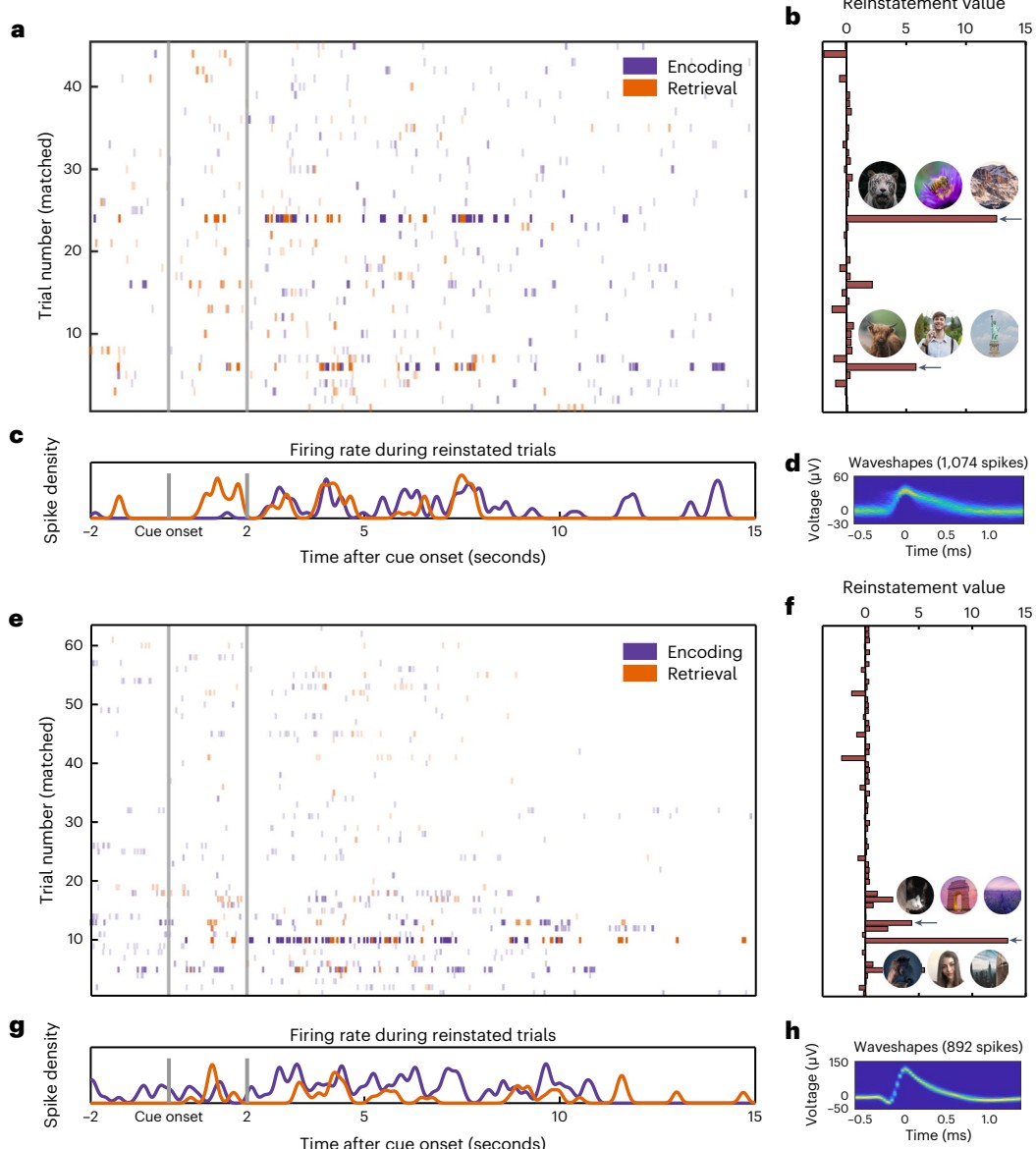

**Fig. 3 | Firing patterns for two example ESNs. a,** Spike raster plot. Each line indicates a spike. Time is shown on the x axis, and episodes are shown on the y axis. The lines are colour-coded in purple for encoding and orange for retrieval. The transparency is adjusted according to the reinstatement values in that specific episode. **b,** Reinstatement values and the animal cues with the respective associate images for reinstated episodes (indicated by the black arrows). **c,** Spike density plot for reinstated episodes. Note that the experiment is self-paced and episode length varies. **d,** 2D histogram of the waveshape of that particular unit[47]. **e–h,** Same as **a–d** but for a different example ESN. The original images have been replaced due to copyright issues.

miss-ESNs when limiting our analysis to later forgotten episodes (15 out of 585 neurons (2.56%); $P = 0.4229$; permutation test). However, this result could stem from a lower number of forgotten events (Supplementary Table 1). To counter this bias, we equalized event numbers between later remembered and later forgotten events for every neuron by randomly sampling (with replacement) later remembered events as many times as participants forgot an event. If any of the sampled events were later reinstated, we considered this neuron a miss-ESN under the null hypothesis. By repeating this procedure 10,000 times, we generated a distribution of how many miss-ESNs were expected if the numbers of later remembered and later forgotten events were equal. This analysis did not result in a significantly lower empirical number of miss-ESNs than hit-ESNs ($P = 0.7032$; bootstrapping test). To conclude, we did not find a significant number of ESNs when restricting our analysis to episodes that were forgotten. However, when considering that fewer episodes

were forgotten than remembered, there was no difference in the number of hit-ESNs and miss-ESNs.

## Identification of temporal ESNs

The previous identification of ESNs relied on a rate code—that is, the standardized mean firing rate during one episode at encoding and retrieval. We have adapted this analysis to identify neurons that reinstate a temporal pattern of firing. For every neuron, we considered the spiking activity from six seconds before until one second after the response during encoding and retrieval (the first and last second were later excluded to avoid edge artefacts).

By convolving each spike with a Gaussian kernel (standard deviation, 100 ms; length, ±300 ms; peak normalized to 1), we created a measure of the instantaneous firing rate. Because we do not know the exact times when an episode is encoded or retrieved, we cross-correlated this episode-specific instantaneous firing rate during

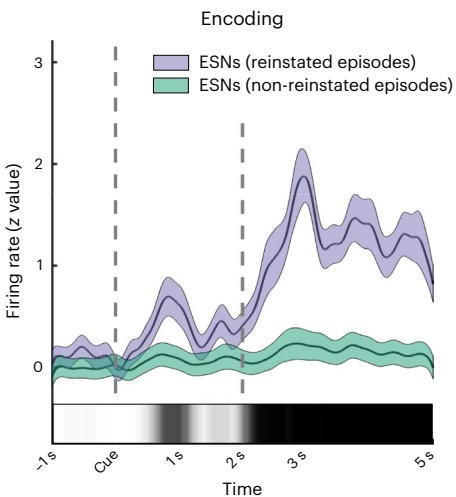
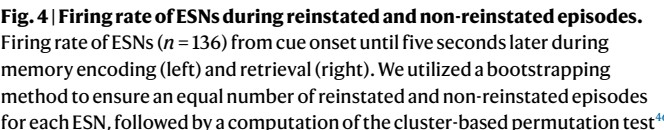
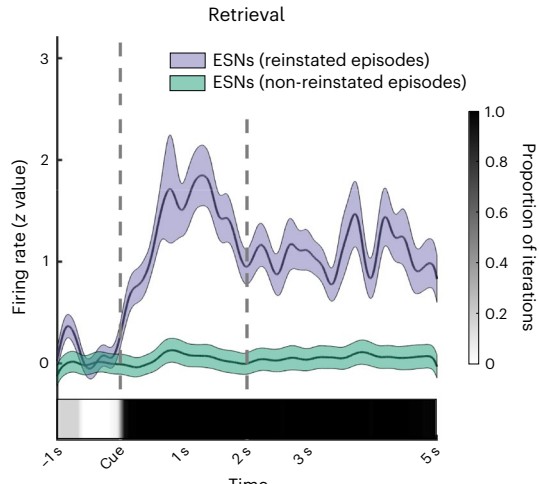

**Fig. 4 | Firing rate of ESNs during reinstated and non-reinstated episodes.** Firing rate of ESNs ($n = 136$) from cue onset until five seconds later during memory encoding (left) and retrieval (right). We utilized a bootstrapping method to ensure an equal number of reinstated and non-reinstated episodes for each ESN, followed by a computation of the cluster-based permutation test[46].

The proportion of iterations that contained a significant cluster at a specific time point is represented by the greyscale inset at the bottom of each plot. The coloured shaded areas represent the s.e.m. that was calculated across all ESNs. We ensured an equal number of reinstated and non-reinstated episodes per ESN while calculating the s.e.m. using a bootstrapping method.

encoding and retrieval and considered the maximum value as the reinstatement value. We repeated this process after shuffling the encoding and retrieval episode order 1,000 times and took the 99th percentile as a threshold for the empirical reinstatement value. If the empirical reinstatement value reached this threshold, we considered the neuron a temporal ESN (tESN; Supplementary Fig. 4). In the next step, we randomly drew for each neuron one of the previously calculated permutations. If these permuted values reached or exceeded the threshold, the neuron was considered a tESN under the null hypothesis. We repeated this process 1,000 times to build a null distribution against which we compared our empirical number of tESNs. We found a significant number of empirical tESNs in Experiment 1 (100 out of 585 neurons (17.09%); $P = 0.016$). For Experiment 2, we further excluded all episodes in which the given neuron showed significant visual tuning (Methods). With this additional constraint, we found a significant number of tESNs in Experiment 2 (40 out of 216 neurons (18.52%); $P < 0.01$).

We employed a permutation test to assess the degree of overlap between episodes reinstated by a rate code (ESNs) and those reinstated by a temporal code (tESNs). Specifically, for each neuron we shuffled the identity of whether an episode was reinstated or not and compared the overlap in the shuffled dataset with the empirical overlap (that is, whether an episode reinstated using a rate code was also reinstated using a temporal code and vice versa). Our analysis revealed a significant overlap in Experiment 1 (Experiment 2), with 20.25% (26.19%) of all episodes reinstated by ESNs also being reinstated by tESNs, and 25.81% (24.44%) of all episodes reinstated by tESNs also being reinstated by ESNs (both $P < 0.001$).

We then tested the validity of this analysis using random spike times. We generated these random spike times by first rounding the empirical spike times to the nearest integer and then drawing an equal number of pseudorandom integer values from a discrete uniform distribution between the first and last empirical spike times. We did not find a significant number of tESNs in either experiment (both $P > 0.2$). We next repeated the analysis using 500 surrogate datasets. These datasets were created by segmenting all spike times into episodes in the order they occurred and then circularly shuffling them. The results of this analysis indicated that the percentage of significant tESN identifications was below the 5% threshold (4.18%), providing further evidence of the credibility and reliability of our analysis.

In conclusion, we show in two separate experiments a significant number of neurons that reinstate an event-specific temporal firing pattern during successful memory retrieval.

### ESNs do not code for time

Recent studies in humans have shown that some hippocampal neurons code specific time points invariant across repetitions, which are referred to as TCs[20,21]. We investigated whether our dataset contains such TCs using a method similar to that employed by Umbach et al.[21]. Due to the self-paced nature of our experiment, each encoding block varied in length. To accommodate this, we used both the unaltered block length and a normalized block length within one recording session (Methods). Of all 585 recorded cells, 12 (normalized) and 10 (non-normalized) fulfilled the criteria of TCs, which is below chance level ($P > 0.9$; permutation test). Critically, there was no significant overlap between neurons that behaved like TCs and ESNs ($P > 0.3$; permutation test).

### ESNs show a wider waveshape than other neurons

We found some evidence that the spike waveshapes of ESNs are wider than those of other units (Supplementary Fig. 5a; $P = 0.0563$ with data from Experiment 1 and $P = 0.0121$ with data from both experiments combined; both independent-samples $t$-test), possibly indicating that ESNs are physiologically different from other neurons. In the hippocampus, a wider waveshape has previously been associated with excitatory cells[26], therefore suggesting that ESNs are predominantly excitatory neurons. There was no significant difference in the spike height or Fano factor between ESNs and other neurons (unpaired $t$-tests; all $P > 0.3$; Supplementary Figs. 5b and 6).

### Neurons are mostly single neurons and not multi-units

Although we tried to separate multi-units into single neurons as well as possible during the spike sorting procedure (Methods), some units might still represent activity from multiple neurons. We thus employed the method outlined by Tankus and colleagues[27] to classify units into single units and multi-units, using the inter-spike interval and spike waveshape variability as objective criteria. In the first experiment, 373 out of 585 units (63.76%) were classified as single units (95/136 ESNs (69.85%)), while in the second experiment, 132 out of 216 units (61.11%) were classified as single units (20/38 ESNs (52.63%)). If we limit

our analysis to neurons that satisfy these stringent criteria for putative single neurons, we still find a significant number of ESNs in the first experiment (95 out of 373 single neurons; $P < 0.001$) and a number of ESNs trending towards significance in the second experiment (21 out of 132 single neurons; $P = 0.0714$).

## Discussion

Using an associative episodic memory paradigm in human epilepsy patients, we identified hippocampal neurons that are active during the initial encoding of a unique episode and later reinstate their firing rate when successfully remembering the same episode. We term these neurons ESNs. The activity of these neurons could not be explained by a firing rate increase towards specific images or time points. These results were replicated across two independent experiments and were robust against variations in how reinstatement was measured as well as changes in the hyperparameter space.

Previous studies have demonstrated that concept neurons increase their firing rate during memory retrieval when the image they are tuned to is part of the memory[10,13]. We used two approaches to ensure that the ESNs we identified are not concept neurons that selectively respond to visual elements or semantic concepts. First, in Experiment 1, we excluded ESNs that were visually responsive to the presentation of the animal cue at encoding. Second, following the episodic memory task in Experiment 2, the participants completed a visual tuning task using all previously presented stimuli. This is a standard method to identify putative concept neurons[7–9,13] and allowed us to exclude episodes where a neuron showed a visual tuning to either the cue or the associate image. Using this approach, we replicated our results from Experiment 1 in a new sample of patients and found a significant number of ESNs while also verifying that these neurons do not selectively respond to visual elements or semantic concepts. Importantly, this finding was robust even when we dramatically reduced the threshold of what constitutes a concept neuron. Taken together, these analyses reinforce the argument that ESNs are memory related.

The existence of ESNs does not exclude concept neurons from playing a role in episodic memory processes. Concept neurons might code the semantic aspect of an episode (that is, the general concept of 'coffee shop'). However, according to the indexing theory[14,15], hippocampal neurons that perform this indexing function should have no initial tuning and are allocated to a specific episode during memory formation (that is, the coffee shop in a specific setting). The behaviour of ESNs would be consistent with such an indexing function and may add crucial event-specific information to an episode, which concept neurons cannot encode themselves.

We found a significant number of ESNs when excluding potential multi-units in the first experiment. However, in the second experiment, we obtained only a statistical trend for a significant number of ESNs. This was probably because this restriction resulted in too few single neurons in the second experiment, thus reducing statistical power.

Because we do not know the exact time points when episodes are encoded or retrieved, we used a rate code approach in the first instance for these analyses (that is, averaging the number of spikes over a time of interest and encoding and retrieval). In addition, we present evidence for a reinstatement of a temporal firing code, which we uncovered by shifting the instantaneous firing rate (that is, the spike times convolved with a Gaussian kernel) using a cross-correlation. Interestingly, we found a significant overlap of episodes that were coded both by tESNs and ESNs. This suggests that in many cases, a temporal firing code can still be identified through a rate code analysis. In some cases, we detected a reinstated episode using either a rate code or a temporal code alone. It is possible that hippocampal neurons employ two distinct coding mechanisms, or we may have missed certain spikes in some instances, resulting in the inability to identify either firing code, which is an interesting question for future studies.

The indexing theory proposes that this coding mechanism is unique to the hippocampus. In line with this, we did not find a significant number of ESNs in the parahippocampus. However, these findings are based on a relatively small sample size and should be considered preliminary. Future studies are needed to ascertain the regional specificity of ESNs to the hippocampus.

We did not find a significant number of ESNs when restricting our analysis to later forgotten episodes. However, there was no significant difference between the number of ESNs when considering later remembered and later forgotten events. This suggests that hippocampal neural reinstatement might occur without behavioural memory retrieval. This could be due to downstream processing being disrupted (that is, due to interference or selective attention). Alternatively, it is possible that in some cases during memory encoding the participants created an episodic memory that did not incorporate the presented associate stimuli. While retrieval would lead to neural reinstatement, the participants would not be able to choose the correct associate images. Another possible explanation for this finding is that a relatively fewer number of forgotten episodes led to an insufficient power to detect a significant number of ESNs that code for later forgotten episodes.

TCs are neurons that invariantly fire at specific, reoccurring time points[20,21]. We did not find a significant number of TCs in our study, and there was no significant overlap between TCs and ESNs. This might be because the self-paced nature of the task introduced too much time variation between too few learning blocks to uncover TC dynamics. However, the absence of TCs in our paradigm corroborates ESNs as independent from TCs.

We found that ESNs have a wider waveshape than other neurons, which suggests that they are physiologically different from other single neurons. Specifically, the broader waveshape of ESNs suggests that they are probably excitatory cells[26]. Alternatively, it is possible that ESNs and neurons with a narrower waveshape are located in different hippocampal subfields. Unfortunately, with the current methods, we lack the precision to designate neurons to individual subfields[28].

One limitation of the current study is that every event was encoded and retrieved only once. However, the very nature of episodic memories is one-shot learning and the ability to subsequently perform mental time travel. Any neural substrate that supports this function must occur after a single bout of learning and subsequent retrieval of a single episode. Our method honours this fundamental characteristic, which is the defining feature of episodic memory as originally stated by Endel Tulving[29]. Arguably, a repeated design would have allowed for a more reliable ESN identification. However, each memory reactivation leads to a transient plasticity of the memory trace until it is reconsolidated again. During this time window, profound changes in the neurons that code for the initial memory trace might occur[30]. To avoid this potential confound, every episode is learned and retrieved only once in the present experiments. The stability of ESNs over repeated reactivations and extended periods therefore remains an interesting topic of research for future studies.

Our results are consistent with previous studies using functional MRI that have shown item-specific activity reinstatement in the hippocampus[31,32] where similar representations are associated with distinct activity patterns[33,34]. These findings are suggestive of an episode-specific neural code, which is consistent with our results. However, due to the coarse resolution of functional MRI, these previous results cannot disambiguate whether this event-specific code is driven by a population of event-specific concept neurons or whether it is driven by a population of event-specific indexing neurons. We here provide evidence for the latter.

Previous intracranial work has identified a multitude of different neurons that detect episode boundaries and event onsets[35] as well as novelty or familiarity[36–39]. Recent work showed that this is a generic signal that can be observed across the brain and is not unique to the

hippocampus[39]. These cell types generally fired not specifically to a particular episode but instead across many episodes[36–38]. Previous research quantified neural firing reinstatement during scene encoding and recognition by considering the activity of all recorded neurons (that is, population activity)[35]. In contrast, the vast majority of ESNs identified in our experiments coded for a single episode, and we demonstrated that neural reinstatement takes place at the single-neuron level. In line with our findings, recent work identified a sparse and item-specific memory recognition signal that was unique to the hippocampus[39]. Importantly, we expect that an episode is coded by an assembly of ESNs from which we sampled only one due to the limited number of neurons that can be recorded with the currently available methods.

Our findings are in line with previous work showing that episodic memories in the hippocampus are coded in a sparse, distributed way[40,41]. However, there are various reasons why we refrain from making any claims regarding sparsity in the present study. A neural code can be sparse in two ways[40]. A neural code can be population sparse, which is the case when a low percentage of neurons respond to a given stimulus. It can also be lifetime sparse, which refers to a low percentage of stimuli that a given neuron responds to. On the one hand, we artificially induced lifetime sparsity (and by extension population sparsity) because we (1) standardized the firing rate during encoding and retrieval and then (2) multiplied these two values. On the other hand, we drastically reduced the sparsity because we tested for reinstatement at each episode without correcting for multiple comparisons. It is very important to understand that while this leads to $\alpha$-level inflation at the level of the neuron, this does not extend to the group level at which we interpret our findings. We have confirmed that our analysis does not have a bias towards positive findings using a simulation (Supplementary Fig. 2). Unfortunately, that also means that in the present study we have to refrain from making any claims regarding lifetime sparsity. Although most ESNs reinstate only a single episode, some code more than one. We expect that episodic memories are represented in the hippocampus as neural assemblies of single neurons and not individual neurons. It is thus plausible that a subset of neurons within a neural assembly coding for memory A are also part of an assembly that codes for memory B. This coding mechanism is more efficient, as a partial overlap between neural assemblies reduces the number of required assemblies.

In conclusion, we found neurons in the hippocampus that show firing reinstatement in response to a specific conjunction of elements within a unique episode. These ESNs did not fire in response to individual concepts (concept neurons) or to specific, reoccurring time points (TCs). We propose that during memory formation an assembly of ESNs acts as a pointer or index that initially binds the elements of an episode together, in line with the indexing theory[14,15,17]. The reactivation of this pointer allows ESNs to reinstate the episodic memory previously encoded. Importantly, because ESNs reinstate unique episodes, they contain a time and content component. However, rather than reflecting the underlying coding mechanism, this time and content aspect necessarily emerges from the conjunctive code of an episode that is unique in content and time.

## Methods
### Procedure of Experiment 1
During the encoding phase of Experiment 1, the participant associated a cue with two other stimuli. For each episode, the cue was a new picture of an animal. The stimuli could be pictures of places, faces of famous persons or both. Every picture was shown only once. Two seconds after the animal cue was presented, the associate stimuli were shown, while the animal cue remained on the screen. The participant was asked to create a vivid imaginary story involving the cue and the two stimuli. This part of the experiment was self-paced. The task continued once the participant rated the plausibility of the imaginary story (plausible/implausible).

After the encoding phase, the participant performed a distractor task to rule out working-memory effects. During the distractor task, the participant had to indicate whether a random number (up to two digits) that appeared serially on the screen was odd or even. After each response, the participant received feedback indicating a correct or incorrect response. This task consisted of 15 trials.

During the retrieval phase, all cues from the previous encoding phase were presented sequentially in pseudorandom order. Each animal cue was presented for two seconds, and the participant was tasked with retrieving the corresponding images. The participant was then asked how many associated images they remembered (none, one or two). The participants had as much time to respond as they required. If the participant indicated that they remembered one or two images, they then were asked to select two pictures from an array of four pictures (two targets and two distractors that consisted of pictures from the previous encoding block, which were associated with a different cue).

The experiment ended after the retrieval phase if the total runtime exceeded 40 minutes or if the participant asked to abort the experiment. Otherwise, the experiment continued with the next encoding block. The encoding block initially consisted of 20 episodes but could be adjusted depending on the cognitive abilities of the participant. If the hit rate fell below 66.25%, fewer episodes were shown for the next block; more episodes were shown if the hit rate surpassed 73.75%.

The participants performed the memory task on a laptop computer (Experiment 1: Toshiba Tecra W50, 60 Hz refresh rate; Experiment 2: Lenovo L390 Yoga, 60.01 Hz refresh rate) while either seated in a chair next to their bed or their hospital bed.

### Procedure of Experiment 2
The second experiment was based on the first experiment with the following adaptations: the participants were presented with one cue image (depicting an animal/place/face of a famous person) and only one associate image (depicting an animal/place/face). During retrieval, the participants were asked whether they remembered the associate image, and they had to choose the correct associate from an array of four pictures (one target and three distractors that consisted of pictures from the previous encoding block, which were associated with a different cue). The experiment was terminated upon request or when the runtime at the end of a retrieval block exceeded 30 minutes.

### Visual tuning task procedure
In Experiment 2, the memory task was followed by a visual tuning task. During this tuning task, every image that was shown during the preceding memory task was displayed. Each image was shown six times in pseudorandom order on the screen for a duration of one second. The inter-image interval was jittered between 500 ms and 550 ms. To ensure attention, the participants had to categorize the image as an animal, a place or a face using the arrow keys on the keyboard.

### Participants
For Experiment 1, eight patients were recorded in the Queen Elizabeth Hospital Birmingham (Birmingham, UK) (4 female; mean age, 36.25 years; from 26 to 49 years) and eight patients in the Universitätsklinikum Erlangen (Erlangen, Germany) (3 female; mean age, 36.125 years; from 26 to 53 years). For Experiment 2, 14 patients were recorded in the Universitätsklinikum Erlangen (Erlangen, Germany) (7 female; mean age, 33.857; from 19 to 58 years). The patients did not receive compensation for their participation.

### Ethical approval
Ethical approval was granted by the National Health Service Health Research Authority (15/WM/0219) and the Ethik-Kommission of the Friedrich-Alexander Universität Erlangen-Nürnberg (142_12 B). Informed consent was obtained in accordance with the Declaration of Helsinki.

## Behavioural analysis

For the analysis of the first experiment, we considered an episode a hit if the participant correctly identified both stimuli. We considered an episode a miss if the participant either indicated no remembrance of any stimuli or did not remember either stimulus correctly. The participants correctly recalled on average 68.38% (s.e. = 4.64%) of the episodes in the first experiment (Supplementary Table 1) and on average 65.63% (s.e. = 4.45%) of the episodes in the second experiment (Supplementary Table 2). This is substantially more than would be expected by chance (16.7% and 25%, respectively).

## Statistical analysis

All statistical analyses were conducted using MATLAB v.R2020a-2022b (MathWorks) on a computer running Windows 10 Enterprise. Furthermore, the following programs were used: Statistics and Machine Learning Toolbox (v.11.7), Fieldtrip (v.20220403), SPM (v.12) and Wave_Clus (v.3). Electrode localization was done with the help of MRIcron (v.1.0.20190902) and visualized using Surf Ice (v.1.0.20201102). The significance threshold for all statistical tests was set at 0.05. Unless specified otherwise, all permutation tests were implemented with $N = 10,000$ random draws.

## Co-registering

For all but one participant, a pre-operational T1-weighted MRI scan was co-registered with a post-operational scan and normalized in Montreal Neurological Institute space using SPM12. For one participant, a post-operational CT scan was used instead of a post-operational MRI scan. Each microelectrode was localized within the hippocampus, within the parahippocampus or outside of both brain structures through visual inspection of an individual MRI (Supplementary Fig. 1). Only activity from microwires in Behnke–Fried electrodes assigned to the hippocampus was analysed in the main analysis of the current study. Neurons in the parahippocampus were analysed in an independent follow-up analysis.

## Recording system and electrodes

The participants were implanted with one to eight (see Supplementary Tables 1 and 2 for an overview) depth electrodes of the Behnke–Fried type with microwire bundles (Ad-Tech Medical Instrument Corporation) to localize epileptic foci. The electrode location was determined by clinical need. These single-use electrodes are made from platinum, have a diameter of 1.3 mm and allow for simultaneous macro- and microcontact recordings. Platinum has a high impedance for lower-frequency and a low impedance for higher-frequency bands. It is therefore suitable for picking up local extracellular action potentials. The microcontacts extended radially past the endpoint of the macro depth electrode, and each contained eight high-impedance microwires (38 μm diameter) and one low-impedance microwire that is typically used for referencing.

The electrodes were connected to an ATLAS system (Neuralynx) running Cheetah software v.1.1.0 (Neuralynx) consisting of CHET-10-A pre-amplifiers and a Digital Lynx NX amplifier and recorded with a sampling rate of either 32,000 Hz (in Birmingham) or 32,768 Hz (in Erlangen). Upon acquisition, an analogue bandpass filter from 0.1 Hz to 9,000 Hz was applied. All experiments were presented using MATLAB v.2015a-2018a (MathWorks) with Psychophysics Toolbox v.3.

## Spike detection and spike sorting

In the following paragraphs, we outline the process used to filter the raw data, detect spike timestamps, extract features of the waveshape and cluster spike waveshapes into putative single neurons using the wave_clus toolbox. For a more in-depth description of the wave_clus algorithm, the reader is referred to Chaure et al[42].

The unfiltered signal included both the local field potential and the action potentials of individual neurons. Action potentials are characterized by a very steep and transient amplitude in the signal. To extract these spikes, we first applied zero-phase filtering using a second-order bandpass elliptic filter in the range of 300–3,000 Hz. The resulting signal contained the information of the so-called spike band.

Next, we segmented the continuous filtered data into epochs of five minutes. Segmenting the continuous data into smaller epochs had the advantage that noise in the signal did not increase the detection threshold for the whole recording and instead was limited to the segment in which it occurred[42].

Spike detection was performed separately for positive and negative deflections. Once a spike was identified, 64 data points around the spike maximum were extracted. This corresponds to a 2 ms window at a sampling rate of 32,000 Hz. The spike peak was aligned to the 20th sampling point. To avoid misalignment of the spike, the waveshape was first up-sampled to 320 data points using cubic spline-interpolated waveforms and then down-sampled again[42].

On the basis of the extracted spike waveform, features were computed using a four-scale multiresolution decomposition with a Haar wavelet. This results in 64-wavelet coefficients for each spike. The ten most significant coefficients were identified using a Lilliefors test and used for the clustering procedure[42].

Non-parametric clustering in the feature space was performed using superparamagnetic clustering. Superparamagnetic clustering grouped spike waves into clusters on the basis of nearest-neighbour interactions[43]. Template-matching in Euclidian space was performed to assign unclassified waveforms to one of the identified clusters. The resulting clustering solution was then manually inspected and further optimized by rejecting artefact clusters, splitting clusters that represented multi-unit activity and merging clusters that probably stemmed from the same neural source. See Supplementary Figs. 5 to 7 for an overview of the spike width, the spike height, the Fano factor and the firing rate separately for ESNs and all other single units.

## Identification of ESNs

For every single unit, we determined the number of spikes within each episode. During encoding, spikes from the onset of the associate images (two seconds after the cue onset—that is, when the whole information of the episode was present) until the end of the episode were considered. During the retrieval phase, spikes from the cue onset until the time point at which the participants indicated how many images they remembered were considered. We chose this time window because an episode could be reinstated following cue presentation, while after the response the participants were presented with an array of images that could have induced single-unit firing. Because the experiment was self-paced and longer episodes trivially contained more spikes, the firing rate (in hertz) was computed for each episode and single unit. In the next step, we z-scored this firing rate per single unit within all encoding episodes and retrieval episodes separately. Afterwards, we excluded all episodes that were later forgotten (for hit-ESNs) or that were later remembered (for miss-ESNs). Only sessions with at least eight episodes after this restriction were considered for further analysis. We then multiplied this standardized firing rate for encoding and retrieval episodes elementwise to gain an indicator for the reinstatement of firing for each episode (Fig. 2a).

To estimate a threshold at which episode-specific firing reinstatement occurs on a single-unit level, we permuted the order of the encoding episodes and recomputed the elementwise product of the shuffled episode series. We repeated this permutation step 10,000 times and stored all output values. The 99th percentile of these pooled values was then used as a threshold for firing reinstatement. As an additional constraint, z-scored firing during encoding and retrieval each had to exceed 1.645 ($P_{\text{right-tailed}} < 0.05$) to make sure the elementwise product was not predominantly driven by a high firing rate in one of the two phases alone (that is, either encoding or retrieval). This procedure allowed us to threshold, but we do

not have family-wise error-corrected statistical significance at the single-unit level (there is no $\alpha$ inflation at the group level; 'Simulation of ESN identification'). Furthermore, we assume that single units fire independently. To ensure that concept neurons tuned to the animal cue were not falsely interpreted as ESN activity, we excluded ESNs that showed a significant firing increase in response to the animal cue at encoding using the method described under 'Identification of putative concept cells'.

Alternative reinstatement measures are explored in the Results under 'Identifying ESNs' and include (1) adding up the standardized firing rates between encoding and retrieval instead of multiplying them (E and R ≥ 1.645; reinstatement, E + R), (2) increasing the minimum standardized firing rate from $z = 1.645$ to $z = 2.6$ (E and R ≥ 2.6; reinstatement, E × R) and (3) using a different reinstatement measure that normalizes the encoding-and-retrieval product by their absolute difference (E and R ≥ 1.645; reinstatement, (E × R)/|E − R|), thereby taking into account the similarity in the standardized firing rate between encoding and retrieval.

In the second step, we calculated whether the number of ESNs (as identified in the above procedure) was above chance level. We did this by randomly choosing one of the permutations calculated in the first step for every single unit and checking whether it would be classified as an ESN under the same criteria outlined above. This approach is similar to a set-level effect in SPM[44]. This process was repeated 10,000 times, and the total number of single units that would be classified as an ESN in every single iteration of this process was used to build a distribution against which we compared our empirically discovered number of ESNs.

### Simulation of ESN identification

We created a simulation using random pseudo-spike rates to determine whether our ESN analysis pipeline contains a bias towards false positive results. To create this simulation, we simulated the firing rate of 585 single neurons during 40 encoding and 40 retrieval trials by randomly drawing from a standard uniform distribution in the open interval of 0 to 1. These values were first multiplied by a variance factor that cycled from 2 to 5 and then $z$-scored independently for encoding and retrieval. Just as in the main ESN analysis, we computed a reinstatement value for each trial by multiplying the two standardized synthetic firing rates. Next, we created a threshold by permuting the encoding and retrieval trial order 10,000 times while recomputing the shuffled reinstatement value. The 99th percentile was used as a threshold, while the empirical standardized pseudo-firing rate had to be at least 1.645 during encoding and retrieval. If these criteria were met, we considered the neuron an ESN.

We then computed the second-order (group-level) permutation test by drawing a random first-order permutation for every single neuron and contrasted these values with the single-neuron-specific threshold. If the shuffled values satisfied the criteria for ESNs (that is, encoding and retrieval standardized pseudo-firing rates at or above 1.645 and a reinstatement value above the neuron-specific threshold), we considered the single neuron an ESN under the null distribution. By repeating this step 10,000 times, we created a distribution under the $H_0$ against which we could compare our initial random values. We repeated this entire process 1,000 times for each level of variance (2 to 5).

Because our initial pseudo-spikes were just random values, we expected 5% of all repetitions to yield a significant number of ESNs at any level of variance. If there was a bias, then more than 5% of all repetitions would contain a significant number of ESNs. As evidenced by Supplementary Fig. 2, this was not the case for any level of variance.

To ensure the robustness of our analysis approach, we repeated the ESN identification analysis using 500 surrogate datasets. These datasets were generated by segmenting all spike times into all available episodes in the order they occurred and then circularly shuffling them.

The results of this analysis revealed that the percentage of significant ESN identifications was 1.8%, which is well below the 5% threshold. These findings provide additional evidence for the credibility and reliability of our analysis.

### Identification of putative concept cells

We followed the method outlined in Mormann et al.[7,8] to detect significant single-unit responses towards images. To this end, the 1,000 ms period after the stimulus onset was divided into 19 overlapping 100 ms bins. The spike counts of each bin over all presentations of an image were compared to the 500 ms baseline periods before stimulus onset for all images in the session using a two-tailed Mann−Whitney $U$-test. We used the Simes procedure to correct for multiple comparisons[45]. We performed this test twice, once with the commonly used threshold of $P < 0.0005$ and again with a liberal threshold of $P = 0.05$.

### Identification of tESNs

The analysis to identify neurons that showed a temporal firing reinstatement for specific episodes closely followed the outline described in 'Identification of ESNs'. For every neuron, we considered the spiking activity from six seconds before until one second after the response during encoding and retrieval (the first and last second were later excluded to avoid edge artefacts). We set a minimum threshold of ten spikes per trial and ten trials per neuron to enter the analysis. These thresholds were chosen to avoid artificially high cross-correlations due to low numbers of spikes, and to have enough trials for the randomization procedure. We then convolved each spike with a Gaussian kernel (standard deviation, 100 ms; length, ±300 ms; peak normalized to 1), creating a measure of instantaneous firing rate.

A main problem with comparing neural time courses between encoding and retrieval is that we do not know the time point at which an episode was encoded or retrieved. We therefore cross-correlated the instantaneous firing rate during encoding with the instantaneous firing rate during the corresponding retrieval trial (maximum lag of ±2.5 s). The maximum value of this sequence served as our empirical reinstatement value. We then shuffled the encoding and retrieval order and recomputed this reinstatement value 1,000 times. The 99th percentile of these values was used as a threshold. If the empirical reinstatement value reached this threshold, we considered the neuron a tESN. In the next step, for each neuron we randomly drew one of the permutations we had calculated previously. Neurons whose permuted values reached or exceeded the threshold were considered tESNs under the null hypothesis. We repeated this process 1,000 times to build a null distribution against which we compared our empirical number of tESNs.

For Experiment 2, we further excluded all trials in which the given neuron showed a significant visual tuning using the methodology outlined under 'Identification of putative concept cells'.

To evaluate the extent of trial overlap between rate code (ESNs) and temporal code (tESNs) reinstated trials, we used a permutation test. The trial identity (reinstated versus non-reinstated) was shuffled within each neuron, and the resulting overlap values were compared with the empirical overlap. The analysis showed a significant overlap between rate code and temporally reinstated episodes in Experiment 1 and Experiment 2 ($P < 0.001$). We tested the validity of this analysis by repeating the same analysis using random spike times. We generated these random spike times by first rounding the empirical spike times to the nearest integer and then drawing an equal number of pseudorandom integer values from a discrete uniform distribution between the first and last empirical spike times. We next followed a similar approach to ensure the robustness of our analysis in the tESN identification as outlined in 'Simulation of ESN identification', whereby we repeated the analysis using 500 surrogate datasets. These datasets were created by segmenting all spike times into episodes in the

order they occurred and then circularly shuffling them. Upon analysis, the results indicated that the percentage of significant tESN identifications was below the 5% threshold (4.18%), providing further evidence of the credibility and reliability of our analysis.

## Spike density calculation

To produce the visualizations in Fig. 4, we extracted spikes from one second before the cue onset until five seconds after the cue onset for each episode. Binary spike times were convolved with a 100 ms Gaussian kernel (length, ±300 ms; peak normalized to 1) to create a time-resolved signal of spike activity. We computed the average firing rate over time for all episodes (ep) during the baseline (BL) period 1,000 ms preceding the animal cue ($\bar{x}_{BL}$). We then z-scored the spike activity during the episode ($x_{ep,t}$) using the standard deviation (s.d. ($\bar{x}_{BL}$)) and mean ($\bar{x}_{BL}$) across all pre-cue baseline periods (equation (1)). To account for instances where no spiking activity occurred during the baseline period, 0.1 (ref. 13) was added to the standard deviation. Episodes were then split into reinstated and non-reinstated episodes. Firing rates for each episode type (reinstated or non-reinstated) were then averaged over ESNs:

$$z_{ep,t} = \frac{x_{ep,t} - \bar{x}_{BL}}{\text{s.d.}(\bar{x}_{BL}) + 0.1} \tag{1}$$

We employed a bootstrapping technique ($N = 100$ random draws) to ensure that the numbers of reinstated and non-reinstated episodes were the same for each ESN. Next, we performed a computation of the s.e.m. and the cluster-based permutation test[46]. The greyscale insets at the bottom of Fig. 4 show the proportion of iterations that had a significant cluster at a specific time point.

## Identification of TCs

We defined the beginning of an encoding block as the most salient event. On the basis of Umbach and colleagues[21], we then extracted all spikes within each block and convolved them with a 251 ms Gaussian kernel (width factor, 2.5). This created a block number × time points matrix. For our first analysis, we cut each encoding block into 40 equally sized bins, thereby normalizing block duration. We then used a Kruskal–Wallis test to determine whether any of the 40 bins significantly differed from each other.

We then performed a circular shifting permutation test to calculate whether we found a significant number of TCs. This is done by shifting a random number of values from the beginning of the vector to the end. This shifting was imposed on each block separately and repeated $N = 10,000$ times for every single unit. In a second test, the block length was determined by the longest block, and shorter blocks were filled up with NaNs values. This resulted in no normalization of time between blocks. The rest of the procedure was the same as described in the previous paragraph.

## Reporting summary

Further information on research design is available in the Nature Portfolio Reporting Summary linked to this article.

## Data availability

The data analysed in this manuscript are available to download on Figshare (https://doi.org/10.6084/m9.figshare.14905317).

## Code availability

The code used in this manuscript is available to download on Figshare (https://doi.org/10.6084/m9.figshare.14905317).

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

## Acknowledgements

This manuscript was supported by the following grants: European Research Council grant no. 647954 (S.H.), European Research Council grant no. 715714 (M.W.) and European Research Council grant no. 101001121 (B.P.S.). Epilepsy Center, Department of Neurology, University Hospital Erlangen, Friedrich-Alexander-Universität Erlangen-Nürnberg, Erlangen, Germany is a Full Member of ERN EpiCare. The funders had no role in study design, data collection and analysis, decision to publish or preparation of the manuscript. We thank all the patients who have participated in our study.

## Author contributions

Conceptualization: L.D.K. and S.H. Data curation: L.D.K. Formal analysis: L.D.K. and S.H. Funding acquisition: S.H., B.P.S. and M.W. Investigation: L.D.K., F.R., G.P., M.T.W., M.V.D.P. and K.W. Project administration: R.C., V.S., D.T.R., J.D.L., S.G., K.W., R.H., G.K. and H.H. Software: L.D.K. Supervision: S.H. and H.B. Visualization: L.D.K., S.H. and M.T.W. Writing—original draft: L.D.K. and S.H. Writing—review and editing: all authors

## Competing interests

The authors declare no competing interests.

## Additional information

**Correspondence and requests for materials** should be addressed to Luca D. Kolibius or Simon Hanslmayr.

# Reporting Summary

## Statistics

For all statistical analyses, confirm that the following items are present in the figure legend, table legend, main text, or Methods section.

| n/a | Confirmed | |
|---|---|---|
| ☐ | ☒ | The exact sample size (*n*) for each experimental group/condition, given as a discrete number and unit of measurement |
| ☐ | ☒ | A statement on whether measurements were taken from distinct samples or whether the same sample was measured repeatedly |
| ☐ | ☒ | The statistical test(s) used AND whether they are one- or two-sided<br>*Only common tests should be described solely by name; describe more complex techniques in the Methods section.* |
| ☒ | ☐ | A description of all covariates tested |
| ☐ | ☒ | A description of any assumptions or corrections, such as tests of normality and adjustment for multiple comparisons |
| ☐ | ☒ | A full description of the statistical parameters including central tendency (e.g. means) or other basic estimates (e.g. regression coefficient) AND variation (e.g. standard deviation) or associated estimates of uncertainty (e.g. confidence intervals) |
| ☐ | ☒ | For null hypothesis testing, the test statistic (e.g. *F*, *t*, *r*) with confidence intervals, effect sizes, degrees of freedom and *P* value noted<br>*Give P values as exact values whenever suitable.* |
| ☒ | ☐ | For Bayesian analysis, information on the choice of priors and Markov chain Monte Carlo settings |
| ☐ | ☒ | For hierarchical and complex designs, identification of the appropriate level for tests and full reporting of outcomes |
| ☒ | ☐ | Estimates of effect sizes (e.g. Cohen's *d*, Pearson's *r*), indicating how they were calculated |

*Our web collection on statistics for biologists contains articles on many of the points above.*

## Software and code

Policy information about availability of computer code

| Data collection | All experiments were presented and analyzed using MATLAB 2015a-2018a (The Mathworks, USA) with Psychophysics Toolbox Version 3 (https://github.com/Psychtoolbox-3/Psychtoolbox-3). Intracranial data was recorded using the ATLAS setup running the Cheetah software v1.1.0 (Neuralynx INC, USA). |
|---|---|
| Data analysis | Data analysis was performed using MATLAB 2020a (The Mathworks, USA) using the Statistics and Machine Learning Toolbox (v11.7), FieldTrip (v20220403), SPM (v12), Wave_clus (v3). Electrode localization was done with the help of MRIcron (v1.0.20190902) and visualized using Surf Ice (v1.0.20201102).<br>Analysis code has been deposited on Figshare (https://figshare.com/s/12fcaf4069e972a5c362). |

For manuscripts utilizing custom algorithms or software that are central to the research but not yet described in published literature, software must be made available to editors and reviewers. We strongly encourage code deposition in a community repository (e.g. GitHub). See the Nature Portfolio guidelines for submitting code & software for further information.

## Data

Policy information about <u>availability of data</u>

All manuscripts must include a <u>data availability statement</u>. This statement should provide the following information, where applicable:

- Accession codes, unique identifiers, or web links for publicly available datasets
- A description of any restrictions on data availability
- For clinical datasets or third party data, please ensure that the statement adheres to our <u>policy</u>

> Data and analysis code is currently available in a private repository on Figshare (https://figshare.com/s/12fcaf4069e972a5c362) and will be made publicly available upon publication. Raw data (.ncs files and MRIs) cannot be shared because they are not anonymized.

## Human research participants

Policy information about <u>studies involving human research participants and Sex and Gender in Research.</u>

| | |
|---|---|
| Reporting on sex and gender | Our first dataset consisted of 16 participants (7 female) while our second dataset consisted of 8 participants (4 female). Sex was not a variable of interest in our research. |
| Population characteristics | All patients had treatment resistant epilepsy and underwent surgery to localize epileptic foci. In our first dataset participants were on average 36.125 years old (26-53 years) while patients in the second dataset were on average 34.375 years old (19-58 years). Age has not been a variable of interest in our research. |
| Recruitment | Patients were admitted into the hospital for localization of epileptic foci using depth electrodes. Epilepsy patients were recruited by the local clinical teams through direct invitations to participate in the experiments. Patients received no componesation for their participation. Patients were not aware of the planned analyses or research questions. All patients gave informed consent and volunteered to participate while hospitalized. |
| Ethics oversight | Ethical approval was granted by the National Health Service Health Research Authority (15/WM/0219) and the Ethik-Kommission of the Friedrich-Alexander Universität Erlangen-Nürnberg (142_12 B). Informed consent was obtained in accordance with the Declaration of Helsinki. |

Note that full information on the approval of the study protocol must also be provided in the manuscript.

# Field-specific reporting

Please select the one below that is the best fit for your research. If you are not sure, read the appropriate sections before making your selection.

☒ Life sciences ☐ Behavioural & social sciences ☐ Ecological, evolutionary & environmental sciences

For a reference copy of the document with all sections, see <u>nature.com/documents/nr-reporting-summary-flat.pdf</u>

# Life sciences study design

All studies must disclose on these points even when the disclosure is negative.

| | |
|---|---|
| Sample size | Our sample size is comparable to other published studies (Zheng et al., 2022 Nat Neurosc [N = 20]; Donoghue et al., 2023 Hippocampus [N = 5]; Kunz et al., 2021, Neuron [N = 15]). No sample-size calculation was performed. |
| Data exclusions | One patient was excluded due to poor/below chance memory performance. Another patient was excluded due to wrong filter settings during data acquisition leading to the loss of spike activity. |
| Replication | We here report findings from two independent datasets. The second dataset included a visual tuning task after a memory task. The memory task in the second experiment was largely the same memory association task as used before, but only consisted of one cue and one associate instead of two associate stimuli. We were able to replicate our original findings in the second dataset. |
| Randomization | All patients performed the same memory association task, so randomization was not applicable here. All covariates (age, sex, number of single neurons) are listed in the supplementary material. |
| Blinding | Patients were unaware of the hypotheses and to be performed data analyses while participating in the experiment. LDK and SH were aware of the hypothesis when analyzing the data. There was no group allocation. |

# Reporting for specific materials, systems and methods

We require information from authors about some types of materials, experimental systems and methods used in many studies. Here, indicate whether each material, system or method listed is relevant to your study. If you are not sure if a list item applies to your research, read the appropriate section before selecting a response.

## Materials & experimental systems

| n/a | Involved in the study |
|-----|------------------------|
| ☒ | ☐ Antibodies |
| ☒ | ☐ Eukaryotic cell lines |
| ☒ | ☐ Palaeontology and archaeology |
| ☒ | ☐ Animals and other organisms |
| ☒ | ☐ Clinical data |
| ☒ | ☐ Dual use research of concern |

## Methods

| n/a | Involved in the study |
|-----|------------------------|
| ☒ | ☐ ChIP-seq |
| ☒ | ☐ Flow cytometry |
| ☒ | ☐ MRI-based neuroimaging |

