## [Peer Review File · Nature Human Behaviour]

Peer Review Information

Journal: Nature Human Behaviour

Manuscript Title: Hippocampal neurons code individual episodic memories in humans

Corresponding author name(s): Luca D. Kolibius and Simon Hanslmayr

Reviewer Comments & Decisions:

Decision Letter, initial version:

26th July 2022

Dear Mr Kolibius,

Thank you once again for your manuscript, entitled "Hippocampal neurons code individual episodic memories in humans" and for your patience during the peer review process.

Your manuscript has now been evaluated by 3 reviewers, whose comments are included at the end of this letter. In the light of their advice, I regret that we cannot offer to publish your manuscript in Nature Human Behaviour.

While the reviewers agree that your work addresses a question of high interest, they all raise concerns about the strength of the novel conclusions that can be drawn at this stage from these data. While Reviewer #3 is more positive than Reviewers #1 and #2, we have carefully considered the reviews as a whole and we feel that many of the concerns expressed would not be addressable given the data available.

I am sorry that we cannot be more positive on this occasion but hope that you will find our reviewers' comments helpful when preparing your paper for submission elsewhere.

Sincerely,
Jamie

Dr Jamie Horder
Senior Editor
Nature Human Behaviour

Reviewers' Comments:

Reviewer #1:

Remarks to the Author:

This study investigated the coding of episodic memory in the human hippocampus using single-unit recordings from epilepsy patients who have had electrodes implanted for clinical monitoring purposes. At encoding, the memory task involved the presentation of a cue (e.g., a picture of tiger) followed by the presentation of another image or two. The participants were instructed to imagine a vivid episode involving the animal cue and the associate image(s), and the activity of hippocampal neurons was recorded during this time. During retrieval, the same cue was presented, and participants were instructed to recall the image(s) that the cue had earlier been associated with. The activity of hippocampal neurons was recorded again during this time. To identify Episode Specific Neurons (ESNs), the normalized spike counts at encoding and retrieval were multiplied together (separately for each neuron), and a neuron was counted as an ESN if (1) the multiplied value fell above the 99th percentile of a bootstrapped (shuffled) distribution and (2) spiking was significantly elevated above baseline at both encoding and retrieval. Using these selection criteria, 141 out of 585 (~24%) hippocampal neurons were classified as ESNs. In Experiment 2, 17 out of 86 (~20%) hippocampal neurons were classified as ESNs. Various strategies were used to rule out the possibility that these neurons did not reflect episodic memory but instead reflected semantic memory (i.e., to rule out the possibility that these were actually concept neurons). The conclusion is that the identified ESNs represent the conjunction of multiple event features.

The question addressed by this study is timely and important, and these are interesting findings. The paper seems like a reasonable candidate for publication, but I am not yet sure about that in light of the four issues I summarize below.

1. The results are certainly consistent with the ESN interpretation, but, as the authors obviously appreciate, the biggest challenge is ruling out the possibility that the putative ESNs are instead concept neurons. In fact, it took me a while to appreciate the likely reason why Experiment 2 was performed and why it included a visual tuning test. If I have it right, the visual tuning test most directly addresses the possibility that ESNs are actually concept neurons that respond to one of the associate images (such as the image of Albert Einstein). In Experiment 1, the following less compelling method used to rule out that possibility:

"To rule out that ESNs are firing in response to a stimulus category (such as faces or places) we repeated the ESN identification analysis, but only included ESNs that showed reinstatement in at least one episode with two face associate images and at least one episode with two place associate images. Such neurons would be very unlikely to code for a specific stimulus category or concept because the stimulus material in these episodes does not overlap. Again, we found a significant number of ESNs (7 out of 585 single units \pm 0.012%; $p = 0.0004$)."

Even if I agree that *these* 7 neurons are not concept neurons, why should the results of this test have any bearing on my interpretation of all of the other ESNs, which still might concept neurons for an associate image? This is why the visual tuning test seems like a better way to go. Using that approach, concept neurons can be identified in the way that they usually are, and then they can be excluded from the analysis. Still, I have two questions about the tuning test that I would like to see

addressed. First, I realize that the tuning test replicated procedures that have been used in prior work to identify concept neurons, but I believe those tests were originally designed to be conservative. That is, in those earlier studies, it was fine to miss a lot of concept neurons to ensure that the identified neurons were true positives. Am I right about that? If so, then it might make sense to use a less conservative test in this new study because a conservative test stacks the deck in favor of the conclusion the authors wish to draw (namely, that they are identifying ESNs, not concept neurons). In supplemental material, it says that to identify putative concept cells, the "spike counts of each bin over all presentations of an image were compared to the 500ms baseline periods before stimulus onset for all images in the session using a two-tailed Mann-Whitney U test ($p < 0.0005$)."

On the surface, that seems like a very conservative test (one that many true concept cells might fail to pass), but perhaps I am thinking about this in the wrong way.

The second question I had about the visual tuning test is why the stimuli apparently failed to trigger episodic memory of having been paired with a cue earlier in the session. If I see the face of Albert Einstein on the visual tuning test, wouldn't I be likely to recall having recently formed a mental image of that face and a tiger (thereby triggering activity in the putative ESN)? But if that happened, the responding ESN would be mistakenly excluded as a concept cell. Taken together, my two concerns about the visual tuning test are that it might be (a) failing to detect true concept cells (because it is a conservative test) while (b) also falsely detecting putative concept cells that are in fact ESNs. A brief consideration of these possibilities would be a useful addition to this paper.

2. The analytical approach to detecting ESNs seems reasonable enough, but it also seems to involve arbitrary decisions that were never justified. For example, why were the normalized spike counts from encoding and retrieval multiplied instead of, say, added together? The answer might be obvious to some, but it was never clear to me (and I have not seen it done before, though perhaps there is ample precedent). Also, in supplemental material, it says "As an additional constraint, z-scored firing during encoding and retrieval each had to exceed 1.645 ($\triangleq p_{\text{right-tailed}} < 0.05$) to make sure the elementwise product was not predominantly driven by a high firing rate in one of the two phases alone (i.e., either encoding or retrieval)." Again, this criterion seems reasonable but arbitrary. Is there some reason to prefer this criterion to a more conservative one (e.g., a z-score of 2.6)?

3. I was surprised by the large number of ESN's that were detected given the sparse coding envisioned by prior computational models of conjunctive neural representations in the hippocampus. Even though an extremely small percentage of hippocampal neurons can be recorded in epilepsy patients, and even though the stimulus set used in this study was not especially large, it was still the case that 20% or more of recorded hippocampal neurons were classified as ESNs. I initially interpreted this result to weigh strongly against either the notion of sparse coding or the notion that these are actually ESNs. However, certain comments in the paper suggested to me that I might be missing something. For example, on page 7, it says:

"It is of note that the proportion of neurons that can be classified as ESNs is proportional to the number of events learned and retrieved (the same is the case for Concept Neurons). As such it is not suitable to determine the sparseness of the hippocampal code" (p. 7).

This makes me think I might be wrong to interpret the results as being inconsistent with sparse coding. But if episodic memories are in fact sparsely coded, why were you able to detect so many ESNs despite only recording from a microscopic fraction of hippocampal neurons? I am probably

missing something obvious here, but I would appreciate a bit more elaboration on this point.

4. Finally, the conjunctive coding model applies to hippocampal neurons in particular. Thus, if these hippocampal neurons are in fact ESNs, one would not expect to find them in other structures, such as the amygdala. If they are instead concept neurons masquerading as ESNs, then, based on prior studies of concept cells, one would expect to find them in the amygdala (and in other structures) as well. Unless I just missed it, nothing is said about a search for ESNs in other brain structures, but it seems to me that something should be said about that. If putative ESNs are selectively detected in the hippocampus, it would greatly strengthen the main theoretical claim of this paper.

Reviewer #2:

Remarks to the Author:

In this manuscript Kolibius et al. propose that human hippocampal neurons encode and reinstate individual episodes. While single-neuron datasets in humans are very rare and this group has an excellent track-record in studying the electrophysiology of human memory, the methodology described in this work is incapable of supporting the conclusions of the manuscript.

A majority of "episode-specific neurons" (ESNs), the core finding of this work, are categorized based on the product of two single-trial mean firing-rate estimates (1 encoding trial X 1 retrieval trial). This presents two fundamental problems.

1. First, and most importantly, extracellular firing rates can be highly variable across trials, even in rodent place cells (PMID: 9501237), the paragon of consistent single-neuron tuning in the hippocampus. It is difficult, if not completely impossible, to assess if these ESN responses are meaningful without repeated observation. I cannot recall of a single study, especially in humans, where a complex process like episodic memory encoding is convincingly presented using only a single trial of mean neuronal spiking.

2. In addition to relying on single-trial estimates of mean firing-rates, the reinstatement-value used to categorize ESN responses is not appropriate, as a product of two values does not imply "reinstatement". The authors themselves note this but still use this flawed metric as the foundation of their permutation testing, with only a weak criterion placed after shuffling to ensure above-average firing rates in encoding and retrieval. It is thus difficult to scrutinize the magnitude of ESN responses, and whether they are greater than those expected by chance.

I make two suggestions for revising the analytical framework of this project to address these critical issues.

1. Because the experimental design necessitates the analysis of single trials, the authors should refrain from collapsing neural activity across time (and thus losing most of the information available to them). Instead, they might explore temporal fluctuations in single-trial firing activity (see PMID: 30006598). This could address the difficulty in interpreting single-trial mean firing rates.

2. An improved reinstatement metric could incorporate the difference between encoding and retrieval

activity (i.e. weight the product by the difference in firing activity: $(E \times R) / |(E-R)|$). In this way, any resulting permutation testing will also take into account this fundamental property of reinstatement and lead to a more interpretable threshold for characterizing cell responses.

This is not an exhaustive list of critiques; I have only raised the critical, foundational issues that preclude proper assessment of this manuscript, and prevent me from recommending it for publication in its current state.

Reviewer #3:

Remarks to the Author:

Overview

The paper titled "Hippocampal neurons code individual episodic memories in humans" sought to test for the existence of Episode Specific Neurons (ESNs) and to distinguish them from previously described Concept Neurons and Time Cells. Working with epilepsy patients undergoing intracranial EEG monitoring, they recorded single neurons in the human hippocampus using implanted microwires. Participants performed 2 distinct, but similar associative encoding and cued retrieval experiments. Analysis of the hippocampal single units demonstrated a set of neurons that respond to a specific set of associated images both at encoding and retrieval, referred to as Episode Specific Neurons. These were distinguished from Concept Neurons that might code for specific elements of the associated set and Time Cells that might fire at specific times during the task across blocks with a variety of analyses. The authors state in the discussion that all of these cell types might work in conjunction during episodic memory retrieval, with ensembles of ESNs acting as an index or pointer for the associated elements of specific events.

This work represents a well-reasoned and critical next step in dissecting the variety of neural codes that might contribute to human episodic memory. The authors utilize thoughtful and creative controls and analyses to uncover the unique nature of ESNs. Using a well-established associative memory paradigm, the authors have produced two complimentary datasets and analyses that have important implications for our understanding of episodic memory. I support the publication of this article in Nature Human Behavior, however a few minor issues should be addressed prior to publication that should help readers best contextualize and understand the interesting findings being presented. I've listed my suggestions for revisions by section and line below. Feel free to ask the editor for me to clarify any of these points if needed. Well done!

Comments

Introduction

1. Are you arguing for the existence of both Concept Neurons and Episode Specific Neurons in paragraph 2? Please clarify.
2. Please refer to the reader to the caption of figure 1A for examples of each type of cell. There is a reference to 1A, but it currently seems to only suggest that ESNs are described in the caption. The operational descriptions of the neuron types in this caption are helpful and might be better included in the introduction. In other words, I found myself initially looking for more direct operational definitions

of each neuron type in the introduction and didn't find it until I read the Fig 1A caption later.

3. It would be helpful to cite or refer to any examples that might differentiate the time it takes to develop a neuron that responds to specific Concepts (concept neurons) vs the hypothesized episodic specific neurons?

4. Indexing theory is not mentioned in the introduction but is used elsewhere in the manuscript to describe the sparsely encoded set of elements within an individual episode that act as pointers to cortical modules during memory reinstatement (Lines 51 and 52, pg 2). Please define this as indexing theory in the introduction.

5. To follow up on my first point, it's currently not entirely clear from the introduction whether concept cells and ESNs are hypothesized to potentially both exist in the hippocampus and work together or that they are mutually exclusive. Is the goal of the paper to adjudicate between these two theories, to establish the existence of ESNs, and/or develop a frame work in which they both might play a role in the episodic memory encoding and retrieval process? Please clarify.

Results

1. Please describe the behavioral results for the memory tasks or refer to the supplemental materials. How successful were participants at remembering the associations between the images and how many trials were available for each condition?

2. How long was the delay in time between the encoding session and the retrieval session? Please describe in the brief procedure description in the main text results.

3. Pg. 7, line 141: While I understand this is a conservative analysis, I'm having trouble understanding how such a low percentage of neurons can be considered a significant number of ESNs. How many were expected due to chance in the permutation tests? Other seemingly similar permutation tests with a higher percentage of neurons out of the 585 are not found to be significant in several other analyses throughout the results. Please clarify.

4. Pg. 9, line 170: How many of the units showed Concept Neuron activity with this visual tuning experiment?

5. Pg. 11, line 198: What was the empirical number of miss-ESNs and hit-ESNs?

6. Interesting analysis of ESNs having wider waveshapes!

7. I also appreciate the more conservative consideration of whether the recorded neurons are single neurons and not multi units. Given that most of the reported ESNs still meet this more stringent criteria for putative single units, do the previously reported results still hold up when the analyses are done just on the 518 or 59 more certain single units?

Discussion

1. Thank you for the thorough and thoughtful consideration and contextualization of the results. Well Done!

Methods

1. Please include a supplemental table or column in the existing tables that lists each patient's seizure onset zones.

2. It might be helpful to describe why the blocks were titrated based on performance from block to block between 66% and 74%.

Minor Comments

1. Line 62, page 3: Typo '. elements'

Following suitable revisions, you may want to consider transferring your manuscript. Although we cannot publish your paper, it may be appropriate for another journal in the Nature Portfolio. If you wish to explore the journals and transfer your manuscript please use our manuscript transfer portal **[LINK REDACTED]. You will not have to re-supply manuscript metadata and files, but please note that this link can only be used once and remains active until used. For more information, please see our manuscript transfer FAQ page.

Note that any decision to opt in to In Review at the original journal is not sent to the receiving journal on transfer. You can opt in to In Review at receiving journals that support this service by choosing to modify your manuscript on transfer. In Review is available for primary research manuscript types only.

Author Rebuttal to Initial comments

For your and the reviewers' convenience this response letter contains:

- 1. The original comments of the reviewers (printed in bold).**
2. Our responses to the reviewers' questions.
3. **Changed sections in the manuscript (highlighted).**

Reviewers' Comments:

Reviewer #1:

Remarks to the Author:

This study investigated the coding of episodic memory in the human hippocampus using single-unit recordings from epilepsy patients who have had electrodes implanted for clinical monitoring purposes. At encoding, the memory task involved the presentation of a

cue (e.g., a picture of tiger) followed by the presentation of another image or two. The participants were instructed to imagine a vivid episode involving the animal cue and the associate image(s), and the activity of hippocampal neurons was recorded during this time. During retrieval, the same cue was presented, and participants were instructed to recall the image(s) that the cue had earlier been associated with. The activity of hippocampal neurons was recorded again during this time. To identify Episode Specific Neurons (ESNs), the normalized spike counts at encoding and retrieval were multiplied together (separately for each neuron), and a neuron was counted as an ESN if (1) the multiplied value fell above the 99th percentile of a bootstrapped (shuffled) distribution and (2) spiking was significantly elevated above baseline at both encoding and retrieval. Using these selection criteria, 141 out of 585 (~24%) hippocampal neurons were classified as ESNs. In Experiment 2, 17 out of 86 (~20%) hippocampal neurons were classified as ESNs. Various strategies were used to rule out the possibility that these neurons did not reflect episodic memory but instead reflected semantic memory (i.e., to rule out the possibility that these were actually concept neurons). The conclusion is that the identified ESNs represent the conjunction of multiple event features.

The question addressed by this study is timely and important, and these are interesting findings. The paper seems like a reasonable candidate for publication, but I am not yet sure about that in light of the four issues I summarize below.

Reviewer #1 Point 1

1. The results are certainly consistent with the ESN interpretation, but, as the authors obviously appreciate, the biggest challenge is ruling out the possibility that the putative ESNs are instead concept neurons. In fact, it took me a while to appreciate the likely reason why Experiment 2 was performed and why it included a visual tuning test. If I have it right, the visual tuning test most directly addresses the possibility that ESNs are actually concept neurons that respond to one of the associate images (such as the image of Albert Einstein). In Experiment 1, the following less compelling method used to rule out that possibility:

“To rule out that ESNs are firing in response to a stimulus category (such as faces or places) we repeated the ESN identification analysis, but only included ESNs that showed reinstatement in at least one episode with two face associate images and at least one episode with two place associate images. Such neurons would be very unlikely to code for a specific stimulus category or concept because the stimulus material in these episodes does not overlap. Again, we found a significant number of ESNs (7 out of 585 single units \triangleq 0.012%;

p = 0.0004)."

Even if I agree that *these* 7 neurons are not concept neurons, why should the results of this test have any bearing on my interpretation of all of the other ESNs, which still might concept neurons for an associate image? This is why the visual tuning test seems like a better way to go. Using that approach, concept neurons can be identified in the way that they usually are, and then they can be excluded from the analysis.

Reviewer #1 Point 1: Authors Answer

We agree with the reviewer that the visual tuning test is a much better way to rule out concept neuron activity. The purpose of the analysis where we showed a significant number of ESNs that fired to both stimulus categories (faces and places) was to see whether our prior ESN findings were representative of a general coding pattern of the hippocampus. We argued that we found this pattern on a limited subset (only considering neurons that reinstate a Face/Face and a Place/Place trial) and that this pattern was sufficiently represented in the ESN activity when considering reinstatement in all trials. As such, these findings would provide further evidence that ESNs are not concept neurons, complementing the second experiment.

However, after adjusting for an error in trigger timing this analysis no longer yields significant results ($p > 0.2$). Only 19 ESNs coded for multiple episodes after fixing this error. Consequently, the number of neurons that were eligible for this analysis (i.e., coded for at least one Face/Face and Place/Place trial) were so few that this analysis wasn't feasible anymore.

In the past, this analysis has only led to confusion and the visual tuning task fulfils the role of ruling out CN much more effectively. We, therefore, decided to remove this analysis from the paper.

Reviewer #1 Point 2

Still, I have two questions about the tuning test that I would like to see addressed. First, I realize that the tuning test replicated procedures that have been used in prior work to identify concept neurons, but I believe those tests were originally designed to be conservative. That is, in those earlier studies, it was fine to miss a lot of concept neurons to ensure that the identified neurons were true positives. Am I right about that? If so, then it might make sense to use a less conservative test in this new study because a conservative test stacks the deck

in favor of the conclusion the authors wish to draw (namely, that they are identifying ESNs, not concept neurons). In supplemental material, it says that to identify putative concept cells, the "spike counts of each bin over all presentations of an image were compared to the 500ms baseline periods before stimulus onset for all images in the session using a two-tailed Mann-Whitney U test ($p < 0.0005$)." On the surface, that seems like a very conservative test

(one that many true concept cells might fail to pass), but perhaps I am thinking about this in the wrong way.

Reviewer #1 Point 2: Authors answer

The reviewer raises the very valid concern that we might be too conservative when applying traditional Concept Neuron detection methods to identify weakly tuned Concept Neurons. We have therefore repeated the concept neuron identification analysis with a p-value cut-off at an uncorrected p-level of 0.05 instead of the previous value of 0.0005. During a typical tuning task an average of 108.7 (min: 80; max: 156) different images are shown and each is tested for visual tuning. There is no correction for multiple testing rendering a threshold at $p < 0.05$ very liberal. Unsurprisingly, this lower threshold resulted in a lot more single neurons being labelled as Concept Neurons tuned to at least one image (155 out of 216). We then repeated the ESN identification pipeline with this new CN exclusion criteria. Remarkably, excluding these very liberally defined CNs only resulted in excluding 2 more ESNs and thus had little bearing on the ESN results (36 ESNs out of 216 neurons, $p = 0.0025$; permutation test).

Changes in text:

Main text p.10 lines 23ff

When excluding Concept Neuron activity in this independent dataset, we replicated our previous results and identified a significant number of ESNs (38 out of 216 neurons $\cong 17.59\%$; $p = 0.0053$; permutation test; Figure 2C). In Experiment 2 34 out of 38 ESNs ($\cong 89.47\%$) coded for a single episode.

However, traditional Concept Neuron detection methods might be too conservative to identify weakly tuned Concept Neurons. To address this concern, we drastically reduced the threshold of what constitutes a Concept Neuron, i.e., lowering the uncorrected threshold from $p = 0.0005$ to $p = 0.05$, which increased the number of Concept Neurons from 58 to 155 (out of 216 neurons). During a typical tuning task an average of 108.7 (min: 80; max: 156) different images are shown and each image is tested for visual tuning. There is no correction for multiple testing rendering a threshold at $p < 0.05$ very liberal.

Remarkably, incorporating this liberal threshold to exclude potential Concept Neurons, had little effect on the number of ESNs which remained almost unchanged (36 out of 216 neurons $\cong 16.67\%$; $p = 0.0025$; permutation test). It is conceivable that some images that are presented during the visual tuning task act as cues that reactivate some ESNs. These reactivated ESNs would then be erroneously rejected as Concept Neurons. However, in practice, only four potential ESNs were excluded based on the visual tuning task (six when lowering the Concept Neuron threshold). We suspect that ESNs were not reactivated during the visual tuning task because the images were shown only for one second. This might be sufficient to cause Concept

Neuron firing, but too short to elicit episodic memory retrieval. Nonetheless, we cannot rule out that in some cases ESNs were reactivated during the visual tuning task and subsequently rejected. However, this would only make our analysis for identifying ESNs more conservative.

Supplements p.7 lines 20ff

We used the Simes' procedure to correct for multiple comparisons (Rødland, 2006). We performed this test twice, once with the commonly used threshold of $p < 0.0005$ and again with a liberal threshold of $p = 0.05$.

Reviewer #1 Point 3

The second question I had about the visual tuning test is why the stimuli apparently failed to trigger episodic memory of having been paired with a cue earlier in the session. If I see the face of Albert Einstein on the visual tuning test, wouldn't I be likely to recall having recently formed a mental image of that face and a tiger (thereby triggering activity in the putative ESN)? But if that happened, the responding ESN would be mistakenly excluded as a concept cell. Taken together, my two concerns about the visual tuning test are that it might be (a) failing to detect true concept cells (because it is a conservative test) while (b) also falsely detecting putative concept cells that are in fact ESNs. A brief consideration of these possibilities would be a useful addition to this paper.

Reviewer #1 Point 3: Authors answer

Excluding ESNs as putative concept neurons due to ESN reactivation during the visual task is a potential issue. However, in practice, there were only four excluded ESNs based on the visual tuning task (at a threshold of $p < 0.0005$; six potential ESNs at a threshold of $p < 0.05$). We suspect that we do not see ESN re-activation during the visual tuning task because images during the visual tuning task are shown only for one second. This is sufficient to cause Concept Neuron firing, but may be too short to elicit episodic memory retrieval. Nonetheless, we cannot rule out that in some cases ESNs re-activated during the visual tuning task and were subsequently rejected. However, this would simply lead to a more conservative analysis, which would stack the deck against us and therefore is acceptable.

Changes in text:

Main text p.11 lines 10ff

Remarkably, incorporating this liberal threshold to exclude potential Concept Neurons, had little effect on the number of ESNs which remained almost unchanged (36 out of 216 neurons \cong 16.67%; $p = 0.0025$; permutation test). It is conceivable that some images that are presented during the visual tuning task act as cues that reactivate some ESNs. These reactivated ESNs would then be erroneously rejected as Concept Neurons. However, in practice, only four

potential ESNs were excluded based on the visual tuning task (six when lowering the Concept Neuron threshold). We suspect that ESNs were not reactivated during the visual tuning task because the images were shown only for one second. This might be sufficient to cause Concept Neuron firing, but too short to elicit episodic memory retrieval. Nonetheless, we cannot rule out that in some cases ESNs were reactivated during the visual tuning task and subsequently rejected. However, this would only make our analysis for identifying ESNs more conservative.

Reviewer #1 Point 4

2. The analytical approach to detecting ESNs seems reasonable enough, but it also seems to involve arbitrary decisions that were never justified. For example, why were the normalized spike counts from encoding and retrieval multiplied instead of, say, added together? The answer might be obvious to some, but it was never clear to me (and I have not seen it done before, though perhaps there is ample precedent). Also, in supplemental material, it says “As an additional constraint, z-scored firing during encoding and retrieval each had to exceed 1.645 ($\cong p_{\text{right-tailed}} < 0.05$) to make sure the elementwise product was not predominantly driven by a high firing rate in one of the two phases alone (i.e., either encoding or retrieval).” Again, this criterion seems reasonable but arbitrary. Is there some reason to prefer this criterion to a more conservative one (e.g., a z-score of 2.6)?

Reviewer #1 Point 4: Authors Answer

We appreciate the reviewers' comments on the hyperparameter space we used to detect ESNs. We are not aware of any other study that demonstrates single neuron firing reinstatement for specific trials, therefore we had to devise a new method. We chose multiplication because it is in line with traditional time-resolved correlation measures and leads to more narrow values (big numbers get bigger, small numbers get smaller). We chose a z-score threshold of 1.645 because 5% seemed like a reasonable trade-off between sensitivity and specificity. The 5% criterion is applied to the encoding AND retrieval standardized firing rate, resulting in a combined threshold of $p = 0.0025$ (i.e., $0.05 * 0.05$) which is more conservative. Nonetheless, it is important to verify that our results are resilient towards reasonable changes in the hyperparameter space. We, therefore, repeated our main ESN analysis using multiple different hyperparameters, such as adding up the firing rates instead of multiplying them (125 ESN; $p < 0.001$), increasing the encoding and retrieval minimum standardized firing rate from $z = 1.645$ to $z = 2.6$ (29 ESN; $p < 0.001$) and using a different measurement that normalizes the encoding and retrieval product by their absolute difference as suggested by Reviewer #2 Point 3 (53 ESN; $p < 0.001$). The main result of the ESN analysis was robust against all these changes in the hyperparameter space.

Changes in text:**Main text p.7 lines 4ff**

Using this approach, we identified a significant number of hippocampal ESNs in Experiment 1 (136 out of 585 neurons \cong 23.25%; $p < 0.001$; permutation test; Figure 2B). Comparable results are obtained when (i) adding up the standardized firing rate between encoding and retrieval instead of multiplying them ($E \& R \geq 1.645$, reinstatement: $E+R$) (125 ESNs; $p < 0.001$), (ii) increasing the minimum standardized firing rate from $z = 1.645$ to $z = 2.6$ ($E \& R \geq 2.6$, reinstatement: $E \cdot R$) (29 ESNs; $p < 0.001$) and (iii) using a different reinstatement measure that normalizes the encoding and retrieval product by their absolute difference ($E \& R \geq 1.645$, reinstatement: $(E \cdot R) / |E - R|$) (53 ESNs; $p < 0.001$). This reinstatement measure has the important advantage of considering the similarity between the encoding and retrieval firing rate. In experiment 1 117 out of 136 ESNs (\cong 86.03%) coded for a single episode. Two example ESNs are shown in Figure 3.

Supplements p.6 lines 6ff

Alternative reinstatement measures are explored in the results section under #Identifying Episode Specific Neurons (ESNs) and include (i) adding up the standardized firing rate between encoding and retrieval instead of multiplying them, (ii) increasing the minimum standardized firing rate from $z = 1.645$ to $z = 2.6$ and (iii) using a different reinstatement measure that normalizes the encoding and retrieval product by their absolute difference (see equation (1)), thereby taking into account the similarity in the standardized firing rate between encoding and retrieval.

Reviewer #1 Point 5

3. I was surprised by the large number of ESN's that were detected given the sparse coding envisioned by prior computational models of conjunctive neural representations in the hippocampus. Even though an extremely small percentage of hippocampal neurons can be recorded in epilepsy patients, and even though the stimulus set used in this study was not especially large, it was still the case that 20% or more of recorded hippocampal neurons were classified as ESNs. I initially interpreted this result to weigh strongly against either the notion of sparse coding or the notion that these are actually ESNs. However, certain comments in the paper suggested to me that I might be missing something. For example, on page 7, it says:

“It is of note that the proportion of neurons that can be classified as ESNs is proportional to the number of events learned and retrieved (the same is the case for Concept Neurons). As such it is not suitable to determine the sparseness of the hippocampal code” (p. 7).

This makes me think I might be wrong to interpret the results as being inconsistent with sparse coding. But if episodic memories are in fact sparsely coded, why were you able to detect so many ESNs despite only recording from a microscopic fraction of hippocampal neurons? I am probably missing something obvious here, but I would appreciate a bit more elaboration on this point.

Reviewer #1 Point 5: Authors Answer

We agree with the reviewers' comments in that we also would expect a sparse neural code in the hippocampus based on previous findings. There are various reasons why we refrain from making any claims regarding sparsity in the present study. A neural code can be sparse in two ways (Wixted et al., 2018; PNAS). A neural code can be population sparse, which is the case when a low percentage of neurons respond to a given stimulus. It can also be lifetime sparse, which refers to a low percentage of stimuli that a given neuron responds to.

On the one hand, we artificially induce lifetime sparsity (and by extension population sparsity) because we (i) standardize the firing rate during encoding and retrieval and then (ii) multiply these two values. On the other hand, we drastically reduce the sparsity because we test for reinstatement at each episode without correcting for multiple comparisons. It is very important to understand that while this leads to alpha-level inflation at the level of the single neuron, this does not extend to the group-level at which we interpret our findings. We have confirmed that our analysis does not have a bias towards positive findings using a simulation (see Figure S2). Unfortunately, that also means that in the present study we have to refrain from making any claims regarding lifetime sparsity.

Changes in text:

Main text p.7 lines 20ff

It is of note that the proportion of neurons that can be classified as ESNs is proportional to the number of events learned and retrieved (the same is the case for Concept Neurons). This is because we apply the threshold derived from the first permutation test to all episodes, without family-wise error correction. As such it is not suitable to determine the sparseness of the hippocampal code. However, the proportion of ESNs of all recorded neurons is useful as an estimation of how many ESNs we can expect in future analyses.

It is crucial to understand that this alpha-level inflation does not extend to the group-level permutation test, where the same number of tests are applied to randomly shuffled data. We have added a simulation using random values as spike rates to show that there is no inflation of the alpha error at the group-level at which we interpret our findings (see Methods; Figure S2).

Main text p.21 lines 19ff

Our findings are in line with previous work showing that episodic memories in the hippocampus are coded in a sparse distributed way (Wixted et al., 2014; Wixted et al. 2018). However, there are various reasons why we refrain from making any claims regarding sparsity in the present study. A neural code can be sparse in two ways (Wixted et al., 2018). A neural code can be population sparse, which is the case when a low percentage of neurons respond to a given stimulus. It can also be lifetime sparse, which refers to a low percentage of stimuli that a given neuron responds to. On the one hand, we artificially induce lifetime sparsity (and by extension population sparsity) because we (i) standardize the firing rate during encoding and retrieval and then (ii) multiply these two values. On the other hand, we drastically reduce the sparsity because we test for reinstatement at each episode without correcting for multiple comparisons. It is very important to understand that while this leads to alpha-level inflation at the level of the neuron, this does not extend to the group-level at which we interpret our findings. We have confirmed that our analysis does not have a bias towards positive findings using a simulation (see Figure A.2). Unfortunately, that also means that in the present study we have to refrain from making any claims regarding lifetime sparsity.

Supplements p.6 lines 25ff

Simulation of ESN identification

We created a simulation using random pseudo spike rates to determine whether our ESN analysis pipeline contains a bias towards false positives. To create this simulation, we simulated the firing rate of 585 single neurons during 40 encoding and 40 retrieval trials by randomly drawing from a standard uniform distribution in the open interval of 0 to 1. These values were first multiplied by a variance factor that cycled from 2 to 5 and then z-scored independently for encoding and retrieval. Just as in the main ESN analysis we computed a reinstatement value for each trial by multiplying the two standardized synthetic firing rates. Next, we created a threshold by permuting the encoding and retrieval trial order 10.000 times while recomputing the shuffled reinstatement value. The 99th percentile was used as a threshold while the empirical standardized pseudo-firing rate had to be at least 1.645 during encoding and retrieval. If these criteria were met, we considered the neuron an ESN.

Then we computed the second order permutation test by drawing a random first order permutation for each single neuron and contrasted these values with the single neuron specific threshold. If the shuffled values satisfied the criteria for ESNs (i.e., encoding and retrieval standardized pseudo-firing rate at or above 1.645 and a reinstatement value above the neuron specific threshold) we considered the single neuron an ESN under the null distribution. By repeating this step 10,000 times we created a distribution under the H0 against which we could compare our initial random values. We repeated this entire process 1,000 times for each level of variance (2 to 5).

Because our initial pseudo-spikes were just random values, we expected 5% of all repetitions to yield a significant number of ESNs at any level of variance. If there was a bias, then more than 5% of all repetitions would contain a significant number of ESNs. As evidenced by Figure S2 this was not the case for any levels of variance.

Figure S2. Simulation of ESN identification.

We created a simulation using random pseudo spike rates to determine whether our ESN analysis pipeline contains a bias towards significant results over multiple levels of variance (x-Axis). For each level of variance, we repeated this step 1,000 times and calculated the proportion of iterations that yield a significant result (y-axis; $p \leq 0.05$). The dotted red line represents the 5%-level, and the straight black line represents the results from the stimulation.

Reviewer #1 Point 6

4. Finally, the conjunctive coding model applies to hippocampal neurons in particular. Thus, if these hippocampal neurons are in fact ESNs, one would not expect to find them in other structures, such as the amygdala. If they are instead concept neurons masquerading as ESNs, then, based on prior studies of concept cells, one would expect to find them in the amygdala (and in other structures) as well. Unless I just missed it, nothing is said about a search for ESNs in other brain structures, but it seems to me that something should be said

about that. If putative ESNs are selectively detected in the hippocampus, it would greatly strengthen the main theoretical claim of this paper.

Reviewer #1 Point 6: Authors Answer

The reviewer raises an excellent point which we addressed by analysing neurons from another region, outside the hippocampus, to test for ESNs. This other region was the parahippocampal cortex, because too few neurons were available for the amygdala. To this end, we repeated our main ESN analysis but limited it to neurons within the parahippocampal cortex. We could not find evidence for ESNs in the parahippocampal cortex (Experiment 1: 15 out of 104 neurons, $p = 0.54$; Experiment 2: 3 out of 25 neurons, $p = 0.12$; both permutation tests). However, while these results are consistent with our hypothesis, some caution is warranted because of the relatively small sample size. In the first experiment all parahippocampal neurons originate from only five different microwire bundles over 14 sessions in five different patients while the findings of the second experiment are based on only 25 neurons. Future studies are needed to further investigate this question.

Changes in text:

Main text p.8 lines 7ff

ESNs are suggested to reflect a unique coding mechanism of the hippocampus (Teyler and DiScenna, 1986; Teyler and Rudy, 2007). In line with this, we did not find a significant number of ESNs in the parahippocampus (15 out of 104 neurons, $p = 0.5396$; permutation test).

However, it should be noted that all 104 parahippocampal neurons originate from only five different microwire bundles over 14 sessions in five different patients, and therefore these results, albeit in line with the indexing theory, should be interpreted with caution. The second experiment did not contain a significant number of ESNs in the parahippocampus (3 out of 25 neurons, $p = 0.1199$).

To conclude, we find a significant number of ESNs in the hippocampus, but not in the parahippocampus. The analysis approach we use to identify ESNs is robust to deviations in the parameter space.

Main text p.19 lines 10ff

The Indexing Theory proposes that this coding mechanism is unique to the hippocampus. In line with this, we did not find a significant number of ESNs in the parahippocampus. However, these findings are based on a relatively small sample size and should be considered preliminary. Future studies are needed to ascertain the regional specificity of ESNs to the hippocampus.

Reviewer #2:**Remarks to the Author:**

In this manuscript Kolibius et al. propose that human hippocampal neurons encode and reinstate individual episodes. While single-neuron datasets in humans are very rare and this group has an excellent track-record in studying the electrophysiology of human memory, the methodology described in this work is incapable of supporting the conclusions of the manuscript.

Reviewer #2 Point 1

A majority of “episode-specific neurons” (ESNs), the core finding of this work, are categorized based on the product of two single-trial mean firing-rate estimates (1 encoding trial X 1 retrieval trial). This presents two fundamental problems.

1. First, and most importantly, extracellular firing rates can be highly variable across trials, even in rodent place cells (PMID: 9501237), the paragon of consistent single-neuron tuning in the hippocampus. It is difficult, if not completely impossible, to assess if these ESN responses are meaningful without repeated observation. I cannot recall of a single study, especially in humans, where a complex process like episodic memory encoding is convincingly presented using only a single trial of mean neuronal spiking.

Reviewer #2 Point 1: Authors Answer

The reviewer raises the important issue of the signal-to-noise ratio that can be acquired with only a single learning and remembering trial. This is a valid and often voiced concern to which our answer is two-fold: (1) The very nature of episodic memories is one-shot learning and the ability to subsequently perform mental time travel. Any neural substrate that supports this function must occur after a single bout of learning and subsequent retrieval of a single episode. Our method honours this fundamental characteristic which is the defining feature of episodic memory as originally stated by Endel Tulving (1972). (2) We base our interpretations not on individual trials, but on the results of the group-level statistics that describe a general pattern of neural firing reinstatement in the brain. It is important to understand that the same single-trial signal is used in the shuffled distributions against which we compare our empirical findings. Therefore, any variance in neural firing, however weak or strong, is fully controlled for and unlikely to contribute to false positive outcomes. We have verified this in a simulation, in which we use random pseudo-spike rates for $N = 585$ single neurons. The spike rates for 40 encoding and retrieval trials are drawn randomly from a standard uniform distribution in the open interval of 0 to 1 and then multiplied by a variance factor that cycled from 2 to 5. We then employed the same ESN identification pipeline that we used for the empirical data. The number of tests that yield a

significant number of ESNs do not exceed 5% at any level of variance (Figure S2). This result clearly demonstrates that our analysis method does not suffer from alpha inflation on the group-level statistics at which we interpret our findings and therefore is entirely valid. Furthermore, our confidence in the validity of our findings is based not only on this sound statistical approach but also on the fact that we have replicated the same findings in an independent dataset.

Changes in text

Main text p.20 lines 13ff

One limitation of the current study is that every event was encoded and retrieved only once. However, the very nature of episodic memories is one-shot learning and the ability to subsequently perform mental time travel. Any neural substrate that supports this function must occur after a single bout of learning and subsequent retrieval of a single episode. Our method honours this fundamental characteristic which is the defining feature of episodic memory as originally stated by Endel Tulving (Tulving, 1972). Arguably, a repeated design would have allowed for a more reliable ESN identification. However, each memory reactivation leads to a transient plasticity of the memory trace until it is reconsolidated again. During this time window profound changes in the neurons that code for the initial memory trace might occur (Nader & Hardt, 2009). To avoid this potential confound, every episode is learned and retrieved only once in the present experiments. The stability of ESNs over repeated reactivations and extended time periods, therefore, remains an interesting topic of research for future studies.

Figure S2. Simulation of ESN identification.

We created a simulation using random pseudo spike rates to determine whether our ESN analysis pipeline contains a bias towards significant results over multiple levels of variance (x-Axis). For each level of variance, we repeated this step 1,000 times and calculated the proportion of iterations that yield a significant result (y-axis; $p \leq 0.05$). The dotted red line represents the 5%-level, and the straight black line represents the results from the stimulation.

Reviewer #2 Point 2

2. In addition to relying on single-trial estimates of mean firing-rates, the reinstatement-value used to categorize ESN responses is not appropriate, as a product of two values does not imply “reinstatement”. The authors themselves note this but still use this flawed metric as the foundation of their permutation testing, with only a weak criterion placed after shuffling to ensure above-average firing rates in encoding and retrieval. It is thus difficult to scrutinize the magnitude of ESN responses, and whether they are greater than those expected by chance.

I make two suggestions for revising the analytical framework of this project to address these critical issues.

1. Because the experimental design necessitates the analysis of single trials, the authors

should refrain from collapsing neural activity across time (and thus losing most of the information available to them). Instead, they might explore temporal fluctuations in single-trial firing activity (see PMID: 30006598). This could address the difficulty in interpreting single-trial mean firing rates.

Reviewer #2 Point 2: Authors Answer

The reviewer makes a good point, in that reinstatement could also be temporal in nature, which indeed is something that we have published on previously (see PMIDs 30944439, 2749460, and 30004850). We therefore followed the reviewer's suggestion and calculated episode specific reinstatement of temporal neural firing patterns. Before explaining this analysis and the results, however, we like to clarify that we chose a rate based code in the first place because of the self-paced nature of our experiment which makes it difficult to precisely identify the time points of encoding and retrieval events within an episode. It was not our intention to indicate in any place of the manuscript that this measure is flawed. The reviewer might have gotten the wrong impression here from a statement that we make on page 6, lines 9-11 of the original manuscript, where we explain why we preferred a rate-based code over a time-based code. Additionally, it's important to consider that when using a standardized firing rate threshold of $z = 1.645$ during encoding and during retrieval, the resulting reinstatement value of 2.706 corresponds to a p-value of $p < 0.00341$ (one-tailed). This suggests that the criterion is not weak. It's worth noting that we were able to replicate these findings using an individual threshold of $z = 2.6$ at both encoding and retrieval (6.76 total).

However, as stated above, we have constructed an analysis method to examine temporal reinstatement of hippocampal neuronal firing patterns as suggested by the reviewer. This analysis builds upon the earlier ESN identification analysis, but instead of a rate code in the form of a standardized firing rate we now convolved each spike with a gaussian kernel (100ms) to estimate the instantaneous firing rate. Because it is impossible to know the exact time when an episode is encoded or retrieved, we cross-correlated this trial specific instantaneous firing rate during encoding and retrieval and considered the maximum value as our reinstatement value. We repeated this process after shuffling the encoding and retrieval trial order 1,000 times and took the 99th percentile as a threshold for the empirical reinstatement value. If the empirical reinstatement value was at least at the permutation-based threshold we consider the neuron a temporal ESN (tESN). Just as before, we next randomly drew one permutation of reinstatement values per neuron and if any was at or above the threshold, we considered it a tESN under the null hypothesis. By repeating this procedure 1,000 times we created a distribution representing how many tESN are expected under the null hypothesis. In experiment 1 we found a significant number of tESN ($p = 0.016$).

In experiment 2, we further excluded all trials in which the given neuron showed a significant visual tuning using the methodology outlined under #Identification of putative Concept Neurons

(supplemental text p.7 lines 10ff). With this additional constraint, we found a significant number of tESN (all $p < 0.01$).

We then tested the validity of these analyses by repeating the same analysis using random spike times. We generated these random spike times by first rounding the empirical spike times to the nearest integer and then drawing an equal number of pseudorandom integer values from a discrete uniform distribution between the first and last empirical spike times. We did not find a significant number of tESN in either experiment (both $p > 0.2$).

Changes in the text

Main text p.13 lines 25ff

Identification of temporal Episode Specific Neurons (tESNs)

The previous identification of ESNs relied on a rate code, i.e., the standardized mean firing rate during one episode at encoding and retrieval. We have adapted this analysis to identify neurons that reinstate a temporal pattern of firing. For every neuron, we considered the spiking activity six seconds before until one second after the response during encoding and retrieval (the first and last second was later excluded to avoid edge artefacts).

By convolving each spike with a gaussian kernel (standard deviation: 100ms, length: length: ± 300 ms, peak normalized to 1) we created a measure of instantaneous firing rate. Because we do not know the exact times when an episode is encoded or retrieved, we cross-correlated this trial-specific instantaneous firing rate during encoding and retrieval and considered the maximum value as the reinstatement value. We repeated this process after shuffling the encoding and retrieval trial order 1,000 times and took the 99th percentile as a threshold for the empirical reinstatement value. If the empirical reinstatement value reached this threshold, we considered the neuron a temporal Episode Specific Neuron (tESN; Figure S4). In the next step, we randomly drew for each neuron one of the previously calculated permutations. If these permuted values reached or exceeded the threshold the neuron was considered a tESNs under the null hypothesis. We repeated this process 1,000 times to build a null distribution against which we compared our empirical number of tESNs. We found a significant number of empirical tESNs in Experiment 1 (100 out of 585 neurons $\hat{=} 17.09\%$; $p = 0.016$). For Experiment 2, we further excluded all trials in which the given neuron showed a significant visual tuning (see Methods). With this additional constraint, we found a significant number of tESN in Experiment 2 (40 out of 216 neurons $\hat{=} 18.52\%$; $p < 0.01$).

We then tested the validity of this analysis using random spike times. We generated these random spike times by first rounding the empirical spike times to the nearest integer and then drawing an equal number of pseudorandom integer values from a discrete uniform distribution between the

first and last empirical spikes times. We did not find a significant number of tESN in either experiment (both $p > 0.2$).

In conclusion, we show in two separate experiments a significant number of neurons that reinstate an event specific temporal firing pattern during successful memory retrieval.

Supplemental text p.7 lines 21ff

Identification of temporal Episode Specific Neurons (tESNs)

The analysis to identify neurons that showed a temporal firing reinstatement for specific episodes closely follows the outline described in #Identification of Episode Specific Neurons (ESNs). For every neuron, we considered the spiking activity six seconds before until one second after the response during encoding and retrieval (the first and last second was later excluded to avoid edge artefacts). We required a minimum of ten spikes per trial and ten trials per neuron to proceed with the analysis.

We then convolved each spike with a gaussian kernel (standard deviation: 100ms, length: ± 300 ms, peak normalized to one) creating a measure of instantaneous firing rate.

The main concern is that we do not know the ground truth of at what time point within a trial an episode was encoded or retrieved. To solve this problem, we cross-correlated the instantaneous firing rate during encoding with the instantaneous firing rate during the corresponding retrieval trial (maximum lag of ± 2.5 s). The maximum value of this sequence served as our empirical reinstatement value. We then shuffled the encoding and retrieval order and recomputed this reinstatement value 1,000 times. The 99th percentile of these values was used as a threshold. If the empirical reinstatement value reached this threshold, we considered the neuron a temporal Episode Specific Neuron (tESN). In the next step, for each neuron we randomly drew one of the permutations we calculated previously. Neurons whose permuted values reached or exceeded the threshold were considered tESNs under the null hypothesis. We repeated this process 1,000 times to build a null distribution against which we compared our empirical number of tESNs.

For experiment 2, we further excluded all trials in which the given neuron showed a significant visual tuning using the methodology outlined under #Identification of putative Concept Neurons.

We tested the validity of this analysis by repeating the same analysis using random spike times.

We generated these random spike times by first rounding the empirical spike times to the nearest integer and then drawing an equal number of pseudorandom integer values from a discrete uniform distribution between the first and last empirical spike times.

Figure S4. Example of a trial reinstated by an example temporal Episode Specific Neuron (tESN).

(A) The original instantaneous firing rate during encoding (blue) and retrieval (orange) 5000ms before the response. Each vertical line represents one action potential.

(B) Same as (A), but with the retrieval firing rate shifted according to the peak in the cross-correlation.

Reviewer #2 Point 3

2. An improved reinstatement metric could incorporate the difference between encoding and retrieval activity (i.e. weight the product by the difference in firing activity: $(E \times R) / |(E-R)|$). In this way, any resulting permutation testing will also take into account this fundamental property of reinstatement and lead to a more interpretable threshold for characterizing cell responses.

This is not an exhaustive list of critiques; I have only raised the critical, foundational issues that preclude proper assessment of this manuscript, and prevent me from recommending it for publication in its current state.

Reviewer #2 Point 3: Authors Answer

We do appreciate the reviewers suggestion to incorporate similarity as an additional factor when searching for ESNs. To this end, we have adapted our analysis pipeline and implemented the suggested alternative reinstatement method where we normalize the standardized encoding and

retrieval firing rates by their absolute difference. Using this approach, we found a significant number of ESNs (53 ESNs; $p < 0.001$).

Although the suggestion to normalise the reinstatement value by the absolute difference between the two seems sensible at first, this first intuition must yield to the problematic convolution of similarity and magnitude. To give an example, a trial with an encoding firing rate of 0.4 and a retrieval firing rate of 0.39 would result in an unreasonably high reinstatement number of 15.6 while a high encoding firing rate of 2 and a high retrieval firing rate of 3 only yields a reinstatement value of 6. Because of this we also included a minimum standardized firing rate of $z = 1.645$ during encoding and retrieval.

Changes in text

Main text p.7 lines 5ff

Using this approach, we identified a significant number of hippocampal ESNs in experiment 1 (136 out of 585 neurons $\pm 23.25\%$; $p < 0.001$; permutation test; Figure 2B). Comparable results are obtained when (i) adding up the standardized firing rate between encoding and retrieval instead of multiplying them ($E \& R \geq 1.645$, reinstatement: $E+R$) (125 ESNs; $p < 0.001$), (ii) increasing the minimum standardized firing rate from $z = 1.645$ to $z = 2.6$ ($E \& R \geq 2.6$, reinstatement: $E * R$) (29 ESNs; $p < 0.001$) and (iii) using a different reinstatement measure that normalizes the encoding and retrieval product by their absolute difference ($E \& R \geq 1.645$, reinstatement: $(E * R) / |E - R|$) (53 ESNs; $p < 0.001$). This reinstatement measure has the important advantage of considering the similarity between the encoding and retrieval firing rate.

Supplements p.5 lines 24ff

Afterwards, we excluded all episodes that were later forgotten (for hit-ESNs) or that were later remembered (for miss-ESNs). We then multiplied this standardized firing rate for encoding and retrieval episodes elementwise to gain an indicator for the reinstatement of firing for each episode (Figure 2A). Alternative reinstatement measures are explored in the result section under #Identifying Episode Specific Neurons (ESNs) and include (i) adding up the standardized firing rate between encoding and retrieval instead of multiplying them, (ii) increasing the minimum standardized firing rate from $z = 1.645$ to $z = 2.6$ and (iii) using a different reinstatement measure that normalizes the encoding and retrieval product by their absolute difference.

Reviewer #3

Remarks to the Author:

Overview

The paper titled “Hippocampal neurons code individual episodic memories in humans” sought to test for the existence of Episode Specific Neurons (ESNs) and to distinguish them from previously described Concept Neurons and Time Cells. Working with epilepsy patients undergoing intracranial EEG monitoring, they recorded single neurons in the human hippocampus using implanted microwires. Participants performed 2 distinct, but similar associative encoding and cued retrieval experiments. Analysis of the hippocampal single units demonstrated a set of neurons that respond to a specific set of associated images both at encoding and retrieval, referred to as Episode Specific Neurons. These were distinguished from Concept Neurons that might code for specific elements of the associated set and Time Cells that might fire at specific times during the task across blocks with a variety of analyses. The authors state in the discussion that all of these cell types might work in conjunction during episodic memory retrieval, with ensembles of ESNs acting as an index or pointer for the associated elements of specific events.

This work represents a well-reasoned and critical next step in dissecting the variety of neural codes that might contribute to human episodic memory. The authors utilize thoughtful and creative controls and analyses to uncover the unique nature of ESNs. Using a well-established associative memory paradigm, the authors have produced two complimentary datasets and analyses that have important implications for our understanding of episodic memory. I support the publication of this article in Nature Human Behavior, however a few minor issues should be addressed prior to publication that should help readers best contextualize and understand the interesting findings being presented. I’ve listed my suggestions for revisions by section and line below. Feel free to ask the editor for me to clarify any of these points if needed. Well done!

Reviewer #3 Point 1

Comments

Introduction

1. Are you arguing for the existence of both Concept Neurons and Episode Specific Neurons in paragraph 2? Please clarify.

Reviewer #3 Point 5

5. To follow up on my first point, it’s currently not entirely clear from the introduction whether concept cells and ESNs are hypothesized to potentially both exist in the hippocampus and work together or that they are mutually exclusive. Is the goal of the

paper to adjudicate between these two theories, to establish the existence of ESNs, and/or develop a frame work in which they both might play a role in the episodic memory encoding and retrieval process? Please clarify.

Reviewer #3 Point 1 & 5: Authors Answer

The reviewer raises an important point that we are happy to clarify. We present evidence for a conjunctive episodic memory code (ESN) that cannot be explained by a content-specific code (CN). This does not mean that the two codes are mutually exclusive. ESNs and CN might work in parallel or tandem to code episodic memories. However, the interplay between the two codes goes beyond the scope of the current manuscript.

Changes in text

Main text p.3 lines 6ff

We leveraged intracranial microwire recordings to investigate the firing patterns of neurons in the human hippocampus and hypothesized that a significant number of hippocampal neurons reinstate their firing rate within a specific episode (i.e., fire during encoding and retrieval). Importantly, these ESNs would code for the conjunctive elements present within an episode and are not tuned to individual elements within the episode. The existence of ESNs does not preclude Concept Neurons from participating in episodic memory processing. However, investigating the role of Concept Neurons in episodic memories goes beyond the scope of this work. As control analyses, we investigated whether this firing activity can be explained by a firing response to specific invariant elements, as occurs in Concept Neurons (Quiroga et al., 2005), or by a time preference, as occurs in Time Cells (TC; Reddy et al., 2021; Umbach et al., 2020).

Reviewer #3 Point 2

2. Please refer to the reader to the caption of figure 1A for examples of each type of cell. There is a reference to 1A, but it currently seems to only suggest that ESNs are described in the caption. The operational descriptions of the neuron types in this caption are helpful and might be better included in the introduction. In other words, I found myself initially looking for more direct operational definitions of each neuron type in the introduction and didn't find it until I read the Fig 1A caption later.

Reviewer #3 Point 2 Authors Answer

We thank the reviewer for this feedback. We have expanded our explanation of what ESNs and CNs are in the introduction and now refer to Figure 1A for both neuron types.

Changes in text

Main text p.2 lines 12ff

One important open question is whether neurons in the hippocampus code for specific elements or an entire episode. Concept Neurons in the hippocampus fire in response to specific invariant elements independent of the context (Gelbard-Sagiv et al., 2008; Mormann et al., 2008; Mormann et al., 2011; Quiroga et al., 2005). One contemporary idea is that the diverse elements that make up an episode are coded by the simultaneous activity of a set of these Concept Neurons (Quiroga, 2012; 2020) or by expanding the selectivity of existing Concept Neurons (Ison et al., 2015). According to this framework when you are sitting in your favourite café with your best friend, one set of Concept Neurons might code for the café and a separate set for your friend (Figure 1A). Alternatively, single units in the hippocampus might sparsely encode a specific set of elements within an individual episode and act as pointers to cortical modules during memory reinstatement.

According to this so-called Indexing Theory (Teyler and DiScenna, 1986; Teyler and Rudy, 2007) a set of hippocampal neurons code for the entire episode (i.e., your friend and the café; Figure 1A). Unlike Concept Neurons, these Episode Specific Neurons (ESNs) would fire in response to the conjunction of all the diverse information within an episode and not in response to individual content elements.

Reviewer #3 Point 3

3. It would be helpful to cite or refer to any examples that might differentiate the time it takes to develop a neuron that responds to specific Concepts (concept neurons) vs the hypothesized episodic specific neurons?

Reviewer #3 Point 3 Authors Answer

The development of Concept Neurons and also the mechanism that gives rise to them make for an exciting but at the same time understudied topic. We are not aware of any studies in this regard. We can conjecture that CN can develop at least on the order of days as some researchers have reported CN tuned to the experimenter (Viskontas et al., 2009), but we are not aware of any published science investigating the time course over which Concept Neurons develop.

Reviewer #3 Point 4

4. Indexing theory is not mentioned in the introduction but is used elsewhere in the manuscript to describe the sparsely encoded set of elements within an individual episode that act as pointers to cortical modules during memory reinstatement (Lines 51 and 52, pg 2). Please define this as indexing theory in the introduction.

Reviewer #3 Point 4 Authors Answer

We thank the reviewer for pointing out this mistake and now properly define the Indexing Theory in the introduction.

Changes in text**Main text p.2 lines 21ff**

Alternatively, single units in the hippocampus might sparsely encode a specific set of elements within an individual episode and act as pointers to cortical modules during memory reinstatement. According to this Indexing Theory (Teyler and DiScenna, 1986; Teyler and Rudy, 2007) a set of hippocampal neurons code for the entire episode (i.e., your friend and the café; Figure 1A).

Reviewer #3 Point 6**Results**

1. Please describe the behavioral results for the memory tasks or refer to the supplemental materials. How successful were participants at remembering the associations between the images and how many trials were available for each condition?

Reviewer #3 Point 6 Authors Answer

We have added the aggregate memory performance to the main text and now refer the reader to the table in the supplemental text.

Changes in text**Main text p.6 lines 5ff**

The experiments were self-paced and every episode was learned and retrieved only once. Participants correctly recalled on average 68.38% ($SE = 4.64\%$) episodes in the first experiment (see Supplements Table S1) and on average 65.63% ($SE = 4.45\%$) episodes in the second experiment (see Supplements Table S2). This is substantially more than would be expected by chance (16.7% and 25% respectively).

Reviewer #3 Point 7

2. How long was the delay in time between the encoding session and the retrieval session? Please describe in the brief procedure description in the main text results.

Reviewer #3 Point 7: Authors Answer

The distractor task consisted of 15 trials and lasted between 22.43s and 224.52s (median duration: 42.19s). We now include that information in the main text.

Changes in text**Main text p.6 lines 1ff**

The encoding and recall phase of the experiment was interleaved with a short distractor task where patients had to judge whether a series of 15 numbers was odd or even. The distractor phase lasted between 22.43s and 224.52s (median duration: 42.19s). During the recall phase, the animal cue was presented again and participants were asked to retrieve the associate image(s).

Reviewer #3 Point 8

3. Pg. 7, line 141: While I understand this is a conservative analysis, I'm having trouble understanding how such a low percentage of neurons can be considered a significant number of ESNs. How many were expected due to chance in the permutation tests? Other seemingly similar permutation tests with a higher percentage of neurons out of the 585 are not found to be significant in several other analyses throughout the results. Please clarify.

Reviewer #3 Point 8: Authors Answer

We thank the reviewer for this helpful comment. The same criteria (i.e., significant reinstatement on at least one trial with two faces as associates and at least one trial with two places as associate images) that we apply for the detection of ESNs on the empirical data we apply on the shuffled data. This means that the number of ESNs we find under the null hypothesis is much smaller than for the unrestricted ESN analysis (i.e., significant reinstatement on any trial). However, after adjusting for an error in trigger timings this analysis no longer yields significant results ($p > 0.2$). This is likely because only 19 ESNs coded for multiple episodes making the analysis not sensitive enough. In the past, this analysis has only led to confusion and the visual tuning task fulfils the role of ruling out CN much more effectively. We, therefore, decided to remove this analysis from the paper.

Reviewer #3 Point 9

4. Pg. 9, line 170: How many of the units showed Concept Neuron activity with this visual tuning experiment?

Reviewer #3 Point 9: Authors Answer

Out of 216 neurons, 56 are labelled as Concept Neurons and show a significant tuning to at least one image. This number increases to 155 Concept Neurons when lowering the Concept Neuron detection threshold from $p = 0.0005$ to $p = 0.05$ (uncorrected). Even when reducing the Concept Neuron detection threshold to $p = 0.05$ we find a significant number of ESNs (36 out of 216 neurons, $p = 0.0025$; permutation test).

Changes in text

Main text p.11 lines 3ff

However, traditional Concept Neuron detection methods might be too conservative to identify weakly tuned Concept Neurons. To address this concern, we drastically reduced the threshold of what constitutes a Concept Neuron, i.e., lowering the uncorrected threshold from $p = 0.0005$ to $p = 0.05$, which increased the number of Concept Neurons from 58 to 155 (out of 216 neurons). During a typical tuning task an average of 108.7 (min: 80; max: 156) different images are shown and each image is tested for visual tuning. There is no correction for multiple testing rendering a threshold at $p < 0.05$ very liberal.

Remarkably, incorporating this liberal threshold to exclude potential Concept Neurons, had little effect on the number of ESNs which remained almost unchanged (36 out of 216 neurons \cong 16.67%; $p = 0.0025$; permutation test). It is conceivable that some images that are presented during the visual tuning task act as cues that reactivate some ESNs. These reactivated ESNs would then be erroneously rejected as Concept Neurons. However, in practice, only four potential ESNs were excluded based on the visual tuning task (six when lowering the Concept Neuron threshold). We suspect that ESNs were not reactivated during the visual tuning task because the images were shown only for one second. This might be sufficient to cause Concept Neuron firing, but too short to elicit episodic memory retrieval. Nonetheless, we cannot rule out that in some cases ESNs were reactivated during the visual tuning task and subsequently rejected. However, this would only make our analysis for identifying ESNs more conservative.

Reviewer #3 Point 10

5. Pg. 11, line 198: What was the empirical number of miss-ESNs and hit-ESNs?

Reviewer #3 Point 10: Authors Answer

Out of 585 hippocampal neurons, there are 136 hit ESNs (p.7 line 5) and 15 miss-ESNs (p. 13 line 12).

Reviewer #3 Point 11

6. Interesting analysis of ESNs having wider waveshapes!

Reviewer #3 Point 11: Authors Answer

Thank you! ESNs expressing different physiological properties than other single neurons has interesting implications. However, we would like to make the reviewer aware that this analysis is reduced to a trend towards significance when limited to data from experiment 1 ($p = 0.0563$; experiment 1 & 2: $p = 0.0108$). These results should therefore be considered with caution.

Changes in text

Main text p.15 lines 19ff

We found some evidence that spike waveshapes of ESNs are wider than those of other units (Figure S5A; $p = 0.0563$; with data from Experiment 1 and $p = 0.0121$ with data from both experiments; both independent samples t-test), possibly indicating that ESNs are physiologically different from other neurons. In the hippocampus, a wider waveshape has previously been associated with excitatory cells (Prestigio et al., 2019), therefore suggesting the ESNs are predominantly excitatory neurons.

Reviewer #3 Point 12

7. I also appreciate the more conservative consideration of whether the recorded neurons are single neurons and not multi units. Given that most of the reported ESNs still meet this more stringent criteria for putative single units, do the previously reported results still hold up when the analyses are done just on the 518 or 59 more certain single units?

Reviewer #3 Point 12: Authors Answer

We thank the reviewer for this suggestion. If we limit our analysis to neurons that satisfy the stringent criteria for putative single neurons, we still find a significant number of ESNs in the first experiment (95 out of 373 single neurons; $p < 0.001$), and a statistical trend for the second experiment (21 out of 132 single neurons; $p = 0.0714$). Although since we first submitted this manuscript we have added neurons to experiment 2, limiting our analysis to single neurons might be too restrictive in the already smaller second experiment.

Changes in text

Main text p.16 lines 11ff

If we limit our analysis to neurons that satisfy these stringent criteria for putative single neurons, we still find a significant number of ESNs in the first experiment (95 out of 373 single neurons; $p < 0.001$), and a number of ESNs trending towards significance in the second experiment (21 out of 132 single neurons; $p = 0.0714$).

Reviewer #3 Point 13

Discussion

1. Thank you for the thorough and thoughtful consideration and contextualization of the results. Well Done!

Reviewer #3 Point 13: Authors Answer

Thank you very much for this positive feedback!

Reviewer #3 Point 14

Methods

1. Please include a supplemental table or column in the existing tables that lists each patient's seizure onset zones.

Reviewer #3 Point 14 Authors Answer

We have added the SOZ to the supplemental tables (see below).

Table S1.

Overview of electrode implantation and memory performance.

Patient ID	Number of sessions	Number of bundles	Number of bundles in hippocampus	Trial number ^a	Hits ^a	Hipp. bundles with SUs ^{a,b}	Number of hipp. SUs ^{a,b}	Number of ESNs ^{a,c}
0002	7	6	6	49.4 (0.4)	43.6 (1.3)	2.6 (0.3)	12.3 (2.9)	2.7 (0.64)
0004	3	4	3	49 (0)	33.7 (0.7)	2 (0)	7.3 (2)	1 (0.58)
0005	4	6	6	49 (0)	41.5 (2.7)	3.3 (0.3)	10.3 (1.5)	3.8 (0.63)
0007	3	4	4	94.3 (1.7)	86.7 (4.9)	4 (0)	16.7 (4.9)	8.7 (2.6)
0008	4	6	4	68.8 (2.3)	39.3 (3.9)	1.8 (0.3)	8.3 (1.6)	1.5 (0.29)
0009	3	8	5	84.3 (7.3)	56.3 (7.3)	4 (0)	17 (1.2)	2.7 (1.2)
0012	4	8	6	53.8 (8.1)	32.5 (6.9)	4 (0.4)	21.5 (4.1)	4.5 (0.87)
0013	6	5	5	76.2 (6.3)	45.3 (8.6)	2.8 (0.3)	17.2 (1.6)	3.7 (1.3)
1003	4	2	1	52.5 (2.4)	46.3 (4.9)	1 (0)	6.3 (0.5)	1.5 (0.87)
1004	2	2	2	84.5 (11.5)	75 (11)	1 (0)	1 (0)	1 (0)
1005	4	2	2	73 (13.9)	33.3 (7)	1.5 (0.3)	3.5 (1)	0.5 (0.29)
1007	5	2	1	49.6 (8.5)	31 (6.3)	1 (0)	4.4 (0.5)	0.8 (0.37)
1008	3	1	1	85 (9.5)	22.3 (6.8)	1 (0)	1.7 (0.7)	0 (0)
1009	2	2	1	82.5 (13.5)	67.5 (12.5)	1 (0)	2.5 (0.5)	0 (0)
1011	2	2	2	51 (10)	38 (5)	1.5 (0.5)	8.5 (0.5)	1.5 (1.5)
1012	3	2	2	39.7 (11.6)	21 (4.9)	1 (0)	7.7 (0.7)	0.67 (0.67)

^a Each number stands for the mean over all experimental sessions with the standard error across sessions in brackets.

^b SUs: Single Units (including ESNs)

^c ESNs: Episode Specific Neurons

^d SOZ: seizure onset zone; hip.: hippocampal; ex.: extrahippocampal; bi: bilaterally

Table S2.

Overview of electrode implantation and memory performance.

Patient ID	Number of sessions	Number of bundles in hippocampus with neurons	Trial number ^a	Hits ^a	Number of hipp. neurons ^{a,b}	Number of ESNs ^{a,c}	SOZ ^d
1013	1	1	49 (0)	37 (0)	16 (0)	3 (0)	ex. (right)
1014	1	1	70 (0)	28 (0)	1 (0)	0 (0)	hip (right)
1015	3	1.7	62 (0.6)	28 (1.8)	9 (0.6)	2.3 (0.3)	hip (right)
1016	3	2	46.7 (4.1)	31.3 (3.8)	17.3 (1.5)	3.3 (0.3)	ex. (bi)
1017	2	1.5	62 (2)	38.5 (0.5)	2 (1)	0 (0)	hip (right)
1018	2	3	56 (9)	31 (5)	16 (1)	1.5 (0.5)	ex. (right)
1019	1	2	64 (0)	49 (0)	12 (0)	5 (0)	ex. (left)
1020	1	2	51 (0)	27 (0)	22 (0)	2 (0)	ex. (right)
1021	1	2	54 (0)	38 (0)	15 (0)	1 (0)	ex. (left)
1022	1	2	62 (0)	28 (0)	10 (0)	2 (0)	hip (right)
1024	1	1	78 (0)	70 (0)	3 (0)	1 (0)	no seizures
1026	1	2	49 (0)	38 (0)	15 (0)	3 (0)	hip (left)
1027	1	2	52 (0)	34 (0)	3 (0)	0 (0)	ex. (left)
1028	1	1	45 (0)	43 (0)	4 (0)	1 (0)	hip (left)

^a Each number stands for the mean over all experimental sessions with the standard error across sessions in brackets.

^b SUs: Single Units (including ESNs)

^c ESNs: Episode Specific Neurons

^d SOZ: seizure onset zone; hip.: hippocampal; ex.: extrahippocampal; bi: bilaterally

Reviewer #3 Point 15

2. It might be helpful to describe why the blocks were titrated based on performance from block to block between 66% and 74%.

Reviewer #3 Point 15 Authors Answer

This was done to make the task not too easy, but still easy enough to avoid frustration on the patient's side. Another consideration was that we wanted to have enough trials for hit-miss analyses (subsequent memory effects; see Roux et al. 2022; eLife; DOI: 10.7554/eLife.78109).

Reviewer #3 Point 16**Minor Comments****1. Line 62, page 3: Typo ‘. elements’****Reviewer #3 Point 16: Authors Answer**

Thank you for pointing this out. Fixed.

References

- Michelmann, S., Bowman, H., & Hanslmayr, S. (2016). The temporal signature of memories: identification of a general mechanism for dynamic memory replay in humans. *PLoS Biology*, *14*(8), e1002528.
- Prestigio, C., Ferrante, D., Valente, P., Casagrande, S., Albanesi, E., Yanagawa, Y., ... & Baldelli, P. (2019). Spike-related electrophysiological identification of cultured hippocampal excitatory and inhibitory neurons. *Molecular Neurobiology*, *56*(9), 6276-6292.
- Roux, F., Parish, G., Chelvarajah, R., Rollings, D. T., Sawlani, V., Hamer, H., ... & Hanslmayr, S. (2022). Oscillations support short latency co-firing of neurons during human episodic memory formation. *Elife*, *11*, e78109.
- Tulving, E. (1972). Episodic and semantic memory. In E. Tulving & W. Donaldson (Eds.), *Organization of memory*. New York: Academic Press.
- Viskontas, I. V., Quiroga, R. Q., & Fried, I. (2009). Human medial temporal lobe neurons respond preferentially to personally relevant images. *Proceedings of the National Academy of Sciences*, *106*(50), 21329-21334.
- Wixted, J. T., Goldinger, S. D., Squire, L. R., Kuhn, J. R., Papesh, M. H., Smith, K. A., ... & Steinmetz, P. N. (2018). Coding of episodic memory in the human hippocampus. *Proceedings of the National Academy of Sciences*, *115*(5), 1093-1098.

Decision Letter, first revision:

30th March 2023

Dear Mr Kolibius,

Thank you once again for your revised manuscript, entitled "Hippocampal neurons code individual episodic memories in humans," and for your patience during the re-review process.

Your manuscript has now been evaluated by the original 3 reviewers. The reviewers found your

manuscript to have significantly improved during revision, and Reviewer #3 is now happy to recommend publication, however you will see that Reviewer #1 and #2 have some remaining points for consideration. We remain very interested in the possibility of publishing your study in Nature Human Behaviour, but would like to consider your response to these outstanding concerns in the form of a revised manuscript before we make a decision on publication.

****In addition to addressing the points of Reviewer #1 and #2, we would also like you to ensure that the code and data, suitable to reproduce the results, are made available for the referees to access in the next round of review.****

Finally, your revised manuscript must comply fully with our editorial policies and formatting requirements. Failure to do so will result in your manuscript being returned to you, which will delay its consideration. If you have any questions about any of our policies or formatting, please don't hesitate to contact me.

In sum, we invite you to revise your manuscript taking into account all reviewer and editor comments. We are committed to providing a fair and constructive peer-review process. Do not hesitate to contact us if there are specific requests from the reviewers that you believe are technically impossible or unlikely to yield a meaningful outcome.

We hope to receive your revised manuscript within 4-8 weeks. I would be grateful if you could contact us as soon as possible if you foresee difficulties with meeting this target resubmission date.

- Include a "Response to the editors and reviewers" document detailing, point-by-point, how you addressed each editor and referee comment. If no action was taken to address a point, you must provide a compelling argument. This response will be used by the editors and reviewers to evaluate your revision.
- Highlight all changes made to your manuscript or provide us with a version that tracks changes.

[REDACTED]

We look forward to seeing the revised manuscript and thank you for the opportunity to review your work. Please do not hesitate to contact me if you have any questions or would like to discuss these revisions further.

Sincerely,

Jamie

Dr Jamie Horder
Senior Editor
Nature Human Behaviour

REVIEWER COMMENTS:

Reviewer #1:
Remarks to the Author:

This revision effectively addressed the concerns I raised in connection with the original version of the paper. First, the dicey test previously used to identify ESNs in Experiment 1 has been dropped, and, second, my concern about the test used in Experiment 2 to identify concept neurons being too conservative was addressed by also using a very liberal test. In fact, the test was so liberal that 155 out of 216 recorded units were classified as concept neurons. They then repeated the ESN identification pipeline with this new concept-neuron exclusion criteria and report the following outcome: "Remarkably, excluding these very liberally defined CNs only resulted in excluding 2 more ESNs and thus had little bearing on the ESN results (36 ESNs out of 216 neurons, $p = 0.0025$; permutation test)."

"Remarkably" is putting it mildly. To me, this outcome is amazing. The authors should double-check this result, but if it holds up, it addresses my concern more convincingly than I thought possible. A puzzle is that one might expect the visual tuning test to reactivate some ESNs, causing them to be incorrectly rejected as concept neurons. Why didn't that happen more often? I asked this question in my previous review. The possible explanation offered by the authors was not very compelling to me, and I think a better possible explanation occurred to me this time. Here is what the authors propose: "We suspect that ESNs were not reactivated during the visual tuning task because the images were shown only for one second. This might be sufficient to cause Concept Neuron firing, but too short to elicit episodic memory retrieval." This is a tad too speculative for my tastes because it is untethered to any prior theoretical or empirical work on episodic memory.

By contrast, the possible explanation I have in mind is tethered to prior work. During the visual tuning task, images from the memory task were repeatedly shown in rapid succession, an approach widely used to identify concept neurons. Unless I just missed it, the instructions that were given to the participants for this task are not reported in the paper, but it does say that this task did not involve a memory component. I therefore assume that the participants were not instructed to use these rapidly presented images to recall what they had earlier been associated with. Here, Endel Tulving's concept of "retrieval mode" would seem to apply. That is, retrieval cues are not like Pavlovian stimuli that automatically elicit a response. Instead, retrieval cues work to the extent that one is in retrieval mode. Retrieval mode is a state of mind that exists when we need to retrieve information. It is characterized by cognitive and neural processes that facilitate the retrieval of episodic memories. Quite a few papers support this idea. Here are two:

Lepage, M., Ghaffar, O., Nyberg, L., & Tulving, E. (2000). Prefrontal cortex and episodic memory retrieval mode. *Proceedings of the National Academy of Sciences of the United States of America*,

97(1), 506–511.

Addante, R. J., Watrous, A. J., Yonelinas, A. P., Ekstrom, A. D., & Ranganath, C. (2011). Prestimulus theta activity predicts correct source memory retrieval. *Proceedings of the National Academy of Sciences of the United States of America*, 108(26), 10702–10707.

In any case, I think a retrieval mode explanation works a lot better than the explanation currently offered by the authors.

I confess to still having slight reservations about the analytical approach used to detect ESNs even though, on the surface, it seems reasonable enough. The method involved multiplying normalized spike counts from encoding and retrieval, and I previously wondered why they were multiplied as opposed to, say, being added. In response, the authors say “We are not aware of any other study that demonstrates single neuron firing reinstatement for specific trials, therefore we had to devise a new method. We chose multiplication because it is in line with traditional time-resolved correlation measures and leads to more narrow values (big numbers get bigger, small numbers get smaller).” This makes sense, but the lack of a formal statistical model backing it up leaves me wondering if there might be some potentially problematic issues with this approach that are hidden from view. But this is only a minor concern for me because the approach does have face validity. And the new information in the paper suggesting that ESNs might be selectively found in the hippocampus (as they should be) helps to validate the entire approach.

Finally, on page 21, it says: “Previous intracranial work has identified a multitude of different neurons that detect novelty or familiarity (Rutishauser et al., 2006; Rutishauser et al., 2008; Rutishauser et al., 2015) as well as episode boundaries and event onsets (Zheng et al., 2022). These cell types generally fired to many episodes, whereas the vast majority of identified ESNs in our experiments coded a single episode. When quantifying neural firing reinstatement between scene encoding and recognition, recent work relied on population activity (i.e., considering the activity of all recorded neurons) (Zheng et al., 2022). In contrast, we showed here that neural reinstatement takes place on the level of a single neuron. Importantly, we expect that an episode is coded by an assembly of ESNs from which we sampled only one due to the limited number of neurons that can be recorded with the currently available methods.”

The distinction between the two types of memory neurons being discussed here is the main focus of a recent paper in PNAS, which might be worth noting:

Urgolites, Z. J., Wixted, J. T., Goldinger, S. D., Papesh, M. H., Treiman, D. M., Squire, L. R., & Steinmetz, P. N. (2022). Two kinds of memory signals in neurons of the human hippocampus. *Proceedings of the National Academy of Sciences*, 119, e2115128119.

Reviewer #2:

Remarks to the Author:

I appreciate the careful job the authors have done addressing my critiques of their methods. The identification of temporal reinstatement, the addition of a normalized reinstatement metric, and the lack of ESNs in parahippocampal cortex have greatly increased my confidence in their findings, and I

think this study will be valuable contribution to studies of human memory electrophysiology. Still, there remain a couple of open questions that should be addressed to sufficiently demonstrate that their unique single-trial reinstatement analysis is not resulting in false-positives:

- I agree with the authors that simulating data, running it through the ESN pipeline, and identifying whether these null data result in significant ESN identification is a good idea for tackling my first critique, in general. However, I am not quite convinced by the execution (the authors simulate firing rates by drawing from a random distribution, which likely does not accurately describe the empirical data). To really be convinced by this control, I would like as much of the empirical spiking to be preserved as possible; to wit, the authors should generate surrogate spike trains by circularly shuffling their empirical spike train data instead, and computed the resulting encoding/retrieval firing rates from these surrogates. This would preserve the underlying spiking statistics of the empirical data while randomizing their relation to the task itself. From this surrogate dataset I would like to see whether the significant number of ESNs identified exceeds 5% when using the same pipeline as the true data.
- The same control described above can be applied to tESNs, rather than drawing spike times from a uniform distribution between the first and last real spike times.
- I suspect Figure 4 is being polluted by a large difference in trial counts between re-instated and non-reinstated episodes. The authors should conduct the same comparison (plot + cluster-permutation test) but with (randomly selected) matched trial counts between conditions.

The following are minor points:

- Does the population of tESNs overlap with ESNs? If not, why not?
- I don't think Figure 4B is ever referenced in the main text.
- Based on the author's revisions, it seems they do not identify multi-episode ESNs any more? If I am interpreting that wrong, then the number of multi-episode ESNs should be noted, and their existence should be discussed with respect to the main theoretical framework.

Reviewer #3:

Remarks to the Author:

The authors have addressed all of my concerns and also seem to have done their best to address the concerns of the other reviewers. Well done!

Author Rebuttal, first revision:

For your and the reviewers' convenience this response letter contains:

1. The original comments of the reviewers (printed in bold).

2. Our responses to the reviewers' questions.

3. Changed sections in the manuscript (highlighted).

Reviewer #1 Point 1

This revision effectively addressed the concerns I raised in connection with the original version of the paper. First, the dicey test previously used to identify ESNs in Experiment 1 has been dropped, and, second, my concern about the test used in Experiment 2 to identify concept neurons being too conservative was addressed by also using a very liberal test. In fact, the test was so liberal that 155 out of 216 recorded units were classified as concept neurons. They then repeated the ESN identification pipeline with this new concept-neuron exclusion criteria and report the following

outcome: “Remarkably, excluding these very liberally defined CNs only resulted in excluding 2 more ESNs and thus had little bearing on the ESN results (36 ESNs out of 216 neurons, $p = 0.0025$; permutation test).”

“Remarkably” is putting it mildly. To me, this outcome is amazing. The authors should double-check this result, but if it holds up, it addresses my concern more convincingly than I thought possible.

Reviewer #1 Point 1: Authors answer

We are glad to hear that our revisions have effectively addressed the reviewer’s concerns raised in the previous version of the paper. We have double-checked the result, and it holds up. Furthermore, we have implemented a code-buddy system in our lab, where members not directly involved in the project check the analysis code for errors and inconsistencies. Additionally, we have made all updated code available online (<https://figshare.com/s/12fc4f4069e972a5c362>), and we encourage anyone to review our code and analysis to verify its accuracy. If there are any errors or issues with our analysis, we would be grateful to know about them, but we are confident in its validity.

Reviewer #1 Point 2

A puzzle is that one might expect the visual tuning test to reactivate some ESNs, causing them to be incorrectly rejected as concept neurons. Why didn't that happen more often? I asked this question in my previous review. The possible explanation offered by the authors was not very compelling to me, and I think a better possible explanation occurred to me this time. Here is what the authors propose: "We suspect that ESNs were not reactivated during the visual tuning task because the images were shown only for one second. This might be sufficient to cause Concept Neuron firing, but too short to elicit episodic memory retrieval." This is a tad too speculative for my tastes because it is untethered to any prior theoretical or empirical work on episodic memory.

By contrast, the possible explanation I have in mind is tethered to prior work. During the visual tuning task, images from the memory task were repeatedly shown in rapid succession, an approach widely used to identify concept neurons. Unless I just missed it, the instructions that were given to the participants for this task are not reported in the paper, but it does say that this task did not involve a memory component. I therefore assume that the participants were not instructed to use these rapidly presented images to recall what they had earlier been associated with. Here, Endel Tulving's concept of "retrieval mode" would seem to apply. That is, retrieval cues are not like Pavlovian stimuli that automatically elicit a response. Instead, retrieval cues work to the extent that one is in retrieval mode. Retrieval mode is a state of mind that exists when we need to retrieve information. It is characterized by cognitive and neural processes that facilitate the retrieval of episodic memories. Quite a few papers support this idea. Here are two:

Lepage, M., Ghaffar, O., Nyberg, L., & Tulving, E. (2000). Prefrontal cortex and episodic memory retrieval mode. *Proceedings of the National Academy of Sciences of the United States of America*, 97(1), 506–511.

Addante, R. J., Watrous, A. J., Yonelinas, A. P., Ekstrom, A. D., & Ranganath,

C. (2011). Prestimulus theta activity predicts correct source memory retrieval. Proceedings of the National Academy of Sciences of the United States of America, 108(26), 10702–10707.

In any case, I think a retrieval mode explanation works a lot better than the explanation currently offered by the authors.

Reviewer #1 Point 2: Authors answer

The reviewer makes an excellent argument in suggesting that patients might not be in a retrieval mode during the visual tuning task, which would explain the lack of ESN firing reinstatement. It is a much more robust argument, and we wholeheartedly agree with it. We have made the necessary adjustments to our manuscript and integrated this argument.

We now also explicitly state the specific instructions given to the participants during the visual tuning task, which involved pressing a designated button to indicate whether the presented image was a face, a place, or an animal.

Changes in text:

Main text p.11 lines 15ff

We suspect that ESNs were not reactivated during the visual tuning task because patients were not instructed to actively retrieve memories and thus may not have been in a "retrieval mode" (Lepage et al., 2000; Addante et al., 2011). Tulving first proposed this concept in 1983 (Tulving, 1983; Nyberg et al., 1995), referring to the cognitive state that occurs when we actively attempt to remember something. Being in a retrieval mode increases the likelihood that a memory cue successfully triggers memory retrieval.

Main text p.25 lines 2f

To ensure attention, patients had to categorize the image as an animal, a place, or a face using the arrow keys on the keyboard.

Reviewer #1 Point 3

I confess to still having slight reservations about the analytical approach used to detect ESNs even though, on the surface, it seems reasonable enough. The method involved multiplying normalized spike counts from encoding and retrieval, and I previously wondered why they were multiplied as opposed to, say, being added. In response, the authors say “We are not aware of any other study that demonstrates single neuron firing reinstatement for specific trials, therefore we had to devise a new method. We chose multiplication because it is in line with traditional time-resolved correlation measures and leads to more narrow values (big numbers get bigger, small numbers get smaller).” This makes sense, but the lack of a formal statistical model backing it up leaves me wondering if there might be some potentially problematic issues with this approach that are hidden from view. But this is only a minor concern for me because the approach does have face validity. And the new information in the paper suggesting that ESNs might be selectively found in the hippocampus (as they should be) helps to validate the entire approach.

Reviewer #1 Point 3: Authors answer

We appreciate the remaining reservations the reviewer has. To address any concerns using the multiplication of firing rates as a reinstatement measure, we added alternative measures such as adding firing rates and normalizing the firing rate product by the absolute difference between firing rates. We furthermore added a simulation that showed that our analysis approach is unbiased. If any concerns remain despite these safeguards, we would like to highlight that our code and data are publicly available for curious readers to access and examine.

Changes in text:**Main text p.15 lines 8ff**

We next repeated the analysis using 500 surrogate datasets. These datasets were created by segmenting all spike times into episodes in the order they occurred, and then circularly shuffling them. The results of this analysis indicated that the percentage of significant tESN identifications was below the 5% threshold

(4.18%), providing further evidence of the credibility and reliability of our analysis.

Main text p.31 lines 13ff

To ensure the robustness of our analysis approach, we repeated the ESN identification analysis using 500 surrogate datasets. These datasets were generated by segmenting all spike times into all available episodes in the order they occurred, and then circularly shuffling them. The results of this analysis revealed that the percentage of significant ESN identifications was 1.8%, which is well below the 5% threshold. These findings provide additional evidence for the credibility and reliability of our analysis.

Main text p.8 lines 2ff

We have added a simulation using random values as spike rates and using circularly shuffled spike times to show that there is no inflation of the alpha error at the group-level at which we interpret our findings (see Methods; Figure S2).

Main text p.33 lines 9ff

We tested the validity of this analysis by repeating the same analysis using random spike times. We generated these random spike times by first rounding the empirical spike times to the nearest integer and then drawing an equal number of pseudorandom integer values from a discrete uniform distribution between the first and last empirical spike times. We next followed a similar approach to ensure the robustness of our analysis in the tESN identification as outlined in #Simulation of ESN identification, whereby we repeated the analysis using 500 surrogate datasets. These datasets were created by segmenting all spike times into episodes in the order they occurred, and then circularly shuffling them. Upon analysis, the results indicated that the percentage of significant tESN identifications was below the 5% threshold (4.18%), providing further evidence of the credibility and reliability of our analysis.

Reviewer #1 Point 4

Finally, on page 21, it says: “Previous intracranial work has identified a multitude of different neurons that detect novelty or familiarity (Rutishauser et al., 2006; Rutishauser et al., 2008; Rutishauser et al., 2015) as well as episode boundaries and event onsets (Zheng et al., 2022). These cell types generally fired to many episodes, whereas the vast majority of identified ESNs in our experiments coded a single episode. When quantifying neural firing reinstatement between scene encoding and recognition, recent work relied on population activity (i.e., considering the activity of all recorded neurons) (Zheng et al., 2022). In contrast, we showed here that neural reinstatement takes place on the level of a single neuron. Importantly, we expect that an episode is coded by an assembly of ESNs from which we sampled only one due to the limited number of neurons that can be recorded with the currently available methods.”

The distinction between the two types of memory neurons being discussed here is the main focus of a recent paper in PNAS, which might be worth noting:

Urgolites, Z. J., Wixted, J. T., Goldinger, S. D., Papesh, M. H., Treiman, D. M., Squire, L. R., & Steinmetz, P. N. (2022). Two kinds of memory signals in neurons of the human hippocampus. *Proceedings of the National Academy of Sciences*, 119, e2115128119.

Reviewer #1 Point 4: Authors answer

We thank the reviewer for bringing the recent work by Urgolites et al. (2022) to our attention. We agree that their study provides valuable insights into the different types of memory neurons in the human hippocampus, and we have incorporated their findings into our discussion of the neural mechanisms underlying memory reinstatement.

Changes in text:

Main text p.21 lines 9ff

Previous intracranial work has identified a multitude of different neurons that detect episode boundaries and event onsets (Zheng et al., 2022) as well as novelty or familiarity (Rutishauser et al., 2006; Rutishauser et al., 2008; Rutishauser et al., 2015, Urgolites et al., 2022). Recent work showed that this is a generic signal that can be observed across the brain and is not unique to the hippocampus (Urgolites et al., 2022). These cell types generally fired not specific to a particular episode, but instead across many episodes (Rutishauser et al., 2006; Rutishauser et al., 2008; Rutishauser et al., 2015). Previous research quantified neural firing reinstatement during scene encoding and recognition by considering the activity of all recorded neurons (i.e., population activity) (Zheng et al., 2022). In contrast, the vast majority of ESNs identified in our experiments coded for a single episode, and we demonstrated that neural reinstatement takes place at the single-neuron level. In line with our findings, recent work identified a sparse and item-specific memory recognition signal that was unique to the hippocampus (Urgolites et al., 2022).

Reviewer #2

I appreciate the careful job the authors have done addressing my critiques of their methods. The identification of temporal reinstatement, the addition of a normalized reinstatement metric, and the lack of ESNs in parahippocampal cortex have greatly increased my confidence in their findings, and I think this study will be valuable contribution to studies of human memory electrophysiology. Still, there remain a couple of open questions that should be addressed to sufficiently demonstrate that their unique single-trial reinstatement analysis is not resulting in false-positives:

Reviewer #2: Authors answer

We appreciate the reviewer's positive feedback and are glad that our efforts to address the raised concerns have increased their confidence in our findings. We are

grateful of their time to review our paper and hope that our additional analyses will address any remaining concerns.

Reviewer #2 Point 1

- I agree with the authors that simulating data, running it through the ESN pipeline, and identifying whether these null data result in significant ESN identification is a good idea for tackling my first critique, in general. However, I am not quite convinced by the execution (the authors simulate firing rates by drawing from a random distribution, which likely does not accurately describe the empirical data). To really be convinced by this control, I would like as much of the empirical spiking to be preserved as possible; to wit, the authors should generate surrogate spike trains by circularly shuffling their empirical spike train data instead, and computed the resulting encoding/retrieval firing rates from these surrogates. This would preserve the underlying spiking statistics of the empirical data while randomizing their relation to the task itself. From this surrogate dataset I would like to see whether the significant number of ESNs identified exceeds 5% when using the same pipeline as the true data.

Reviewer #2 Point 1: Authors answer

We appreciate the reviewer's suggestion to use circularly shuffled data instead of a random distribution to more accurately reflect the empirical data. We agree that this is a more rigorous approach, and we have now repeated the analysis using 500 surrogate datasets. These datasets were generated by segmenting all spike times into all available episodes, in the order they occurred, and then circularly shuffling them. The results of the new analysis demonstrate that the percentage of significant ESN identifications was 1.8% (4.18% for tESNs), which did not exceed the 5% threshold. This provides further support for the validity of our approach and the reliability of our results. We have updated the manuscript to reflect this new analysis and included the relevant statistics in the supplementary materials.

Changes in text:**Main text p.15 lines 8ff**

We next repeated the analysis using 500 surrogate datasets. These datasets were created by segmenting all spike times into episodes in the order they occurred, and then circularly shuffling them. The results of this analysis indicated that the percentage of significant tESN identifications was below the 5% threshold (4.18%), providing further evidence of the credibility and reliability of our analysis.

Main text p.31 lines 13ff

To ensure the robustness of our analysis approach, we repeated the ESN identification analysis using 500 surrogate datasets. These datasets were generated by segmenting all spike times into all available episodes in the order they occurred, and then circularly shuffling them. The results of this analysis revealed that the percentage of significant ESN identifications was 1.8%, which is well below the 5% threshold. These findings provide additional evidence for the credibility and reliability of our analysis.

Reviewer #2 Point 2

- The same control described above can be applied to tESNs, rather than drawing spike times from a uniform distribution between the first and last real spike times.

Reviewer #2 Point 2: Authors answer

See our answer to Reviewer #2 Point 1. We have updated our analysis approach and did not identify a bias in our approach.

Changes in text:**Main text p.8 lines 2ff**

We have added a simulation using random values as spike rates and using circularly shuffled spike times to show that there is no inflation of the alpha error at the group-level at which we interpret our findings (see Methods; Figure S2).

Main text p.33 lines 9ff

We tested the validity of this analysis by repeating the same analysis using random spike times. We generated these random spike times by first rounding the empirical spike times to the nearest integer and then drawing an equal number of pseudorandom integer values from a discrete uniform distribution between the first and last empirical spike times. We next followed a similar approach to ensure the robustness of our analysis in the tESN identification as outlined in #Simulation of ESN identification, whereby we repeated the analysis using 500 surrogate datasets. These datasets were created by segmenting all spike times into episodes in the order they occurred, and then circularly shuffling them. Upon analysis, the results indicated that the percentage of significant tESN identifications was below the 5% threshold (4.18%), providing further evidence of the credibility and reliability of our analysis.

Reviewer #2 Point 3

- I suspect Figure 4 is being polluted by a large difference in trial counts between re-instated and non-reinstated episodes. The authors should conduct the same comparison (plot + cluster-permutation test) but with (randomly selected) matched trial counts between conditions.

Reviewer #2 Point 3: Authors answer

We thank the reviewer for their feedback regarding Figure 4. We agree that the difference in trial counts between reinstated and non-reinstated episodes could have an impact on the results. To address this issue, we employed a bootstrapping approach, where we repeatedly ($N = 100$) drew random samples to equalize the number of reinstated and non-reinstated episodes for each ESN. We then recalculated the SEM and the cluster-based permutation test. The results of the latter are presented as an inset at the bottom of the figure, where timepoints with a higher proportion of bootstrap iterations showing significant differences are represented by more black shading. We hope that this additional analysis provides

a more accurate representation of the data, and we appreciate the reviewer's helpful feedback.

Changes in text:

Main text p. 17, lines 1ff

Figure 4. Firing rate of ESNs during reinstated (purple) and non-reinstated (green) episodes.

Firing rate of ESNs ($n = 136$) from cue onset until five seconds later during memory encoding (left) and retrieval (right). We utilized a bootstrapping method to ensure an equal number of reinstated and non-reinstated episodes for each ESN, followed by a computation of the cluster-based permutation test (Maris and Oostenveld, 2007). The proportion of iterations that contained a significant cluster at a specific timepoint is represented by the grayscale inset at the bottom of the figure. The colored shaded areas represent the standard error of the mean (SEM) that was calculated across all ESNs. We ensured an equal number of reinstated and non-reinstated episodes per ESN while calculating the SEM using a bootstrapping method.

Main text p.34 lines 11ff

We employed a bootstrapping technique ($N = 100$ random draws) to ensure that the number of reinstated and non-reinstated episodes was the same for each ESN. Next, we performed a computation of the standard error of the mean (SEM) and the cluster-based permutation test (Maris and Oostenveld, 2007). The grayscale inset at the bottom of the figure shows the proportion of iterations that had a significant cluster at a specific time point.

Reviewer #2 Point 4

- Does the population of tESNs overlap with ESNs? If not, why not?

Reviewer #2 Point 4: Authors answer

The reviewer suggests an interesting analysis regarding the overlap between the populations of tESNs and ESNs, which we adapted slightly. We would like to clarify our approach and reasoning. The reviewer inquired about the overlap in populations between tESNs and ESNs which would be straight forward to compute, but may not be very informative because a given neuron (qualifying as ESN and tESN) might code for one episode as tESN, and for another as ESN. Therefore, we opted to compute an overlap in terms of the episodes that tESNs and ESNs code.

To investigate an overlap between episodes that were reinstated by ESNs and tESNs, we utilized a permutation test. By shuffling the trial reinstatement status for each neuron, we could compare the empirical overlap with the shuffled reinstated trials. In Experiment 1 (Experiment 2), our analysis revealed a significant overlap between ESNs and tESNs. Specifically, 20.25% (26.19%) of all episodes reinstated by ESNs were also reinstated by tESNs. Additionally, 25.81% (24.44%) of all episodes reinstated by tESNs were also reinstated by ESNs. Both overlap percentages yielded p-values less than 0.001 when compared to the surrogate dataset. This suggests that in many cases, a temporal firing code can still be identified through a rate code analysis. In some cases we detected a reinstated episode using either a rate code or a temporal code alone. It is possible that hippocampal neurons employ two distinct coding mechanisms, or in some

instances, we may have missed certain spikes, resulting in the inability to identify either firing code. This would be an interesting question for future studies to address.

Changes in text:

Main text p.14 lines 21ff

We employed a permutation test to assess the degree of overlap between trials reinstated by a rate code (ESNs) and those reinstated by a temporal code (tESNs). Specifically, for each neuron we shuffled the identity of whether an episode was reinstated or not and compared the overlap in the shuffled dataset with the empirical overlap (i.e., is an episode reinstated using a rate code also reinstated using a temporal code and vice versa). Our analysis revealed a significant overlap in Experiment 1 (Experiment 2), with 20.25% (26.19%) of all trials reinstated by ESNs also being reinstated by tESNs, and 25.81% (24.44%) of all trials reinstated by tESNs also being reinstated by ESNs (both $p < 0.001$).

Main text p.14 lines 9ff

In addition, we present evidence for a reinstatement of a temporal firing code which we uncovered by shifting the instantaneous firing rate (i.e., the spike times convolved with a gaussian kernel) using a cross-correlation. This suggests that in many cases, a temporal firing code can still be identified through a rate code analysis. In some cases we detected a reinstated episode using either a rate code or a temporal code alone. It is possible that hippocampal neurons employ two distinct coding mechanisms, or in some instances, we may have missed certain spikes, resulting in the inability to identify either firing code, which is an interesting question for future studies to address.

Reviewer #2 Point 5

- I don't think Figure 4B is ever referenced in the main text.

Reviewer #2 Point 5: Authors answer

We thank the reviewer for pointing this out. We deleted the letter indices in Figure 4 and now only index Figure 4.

Reviewer #2 Point 6

- Based on the author's revisions, it seems they do not identify multi-episode ESNs any more? If I am interpreting that wrong, then the number of multi-episode ESNs should be noted, and their existence should be discussed with respect to the main theoretical framework.

Reviewer #2 Point 6: Authors answer

In the current version of the manuscript, we only indirectly include information regarding multi-episode ESNs. Specifically, we report the percentages of ESNs that coded for a single episode in Experiment 1 (p. 7, line 15): "In Experiment 1, 117 out of 136 ESNs (86.03%) coded for a single episode," and in Experiment 2 (p. 11, lines 1f): "In Experiment 2, 34 out of 38 ESNs (89.47%) coded for a single episode.". We now point this out explicitly by adding "[...], whereas the rest coded for multiple episodes."

We expect that episodic memories are represented in the hippocampus as neural assemblies and not individual neurons. Thus, it is plausible, that a subset of neurons within a neural assembly coding for memory A are also part of an assembly that codes for memory B. This coding mechanism is more efficient as a partial overlap between neural assemblies reduces the number of required assemblies.

Changes in text:

Main text p.7 lines 15f

In Experiment 1, 117 out of 136 ESNs ($\cong 86.03\%$) coded for a single episode, whereas the rest coded for multiple episodes.

Main text p.16 lines 1f

In Experiment 2, 34 out of 38 ESNs ($\cong 89.47\%$) coded for a single episode, whereas the rest coded for multiple episodes.

Main text p.22 lines 13ff

Although most ESNs reinstate only a single episode, some code more than one. We expect that episodic memories are represented in the hippocampus as neural assemblies of single neurons and not individual neurons. Thus, it is plausible, that a subset of neurons within a neural assembly coding for memory A are also part of an assembly that codes for memory B. This coding mechanism is more efficient as a partial overlap between neural assemblies reduces the number of required assemblies.

Reviewer #3:

The authors have addressed all of my concerns and also seem to have done their best to address the concerns of the other reviewers. Well done!

Reviewer #3 Authors Answer

We thank the reviewer for taking the time to assess our work and providing us with their positive feedback. We are thrilled to hear that we were able to address all of their concerns and those of the other reviewers.

Decision Letter, second revision:

20th June 2023

Dear Dr. Kolibius,

Thank you for your patience as we've prepared the guidelines for final submission of your Nature Human Behaviour manuscript, "Hippocampal neurons code individual episodic memories in humans" (NATHUMBEHAV-22061399B). Please carefully follow the step-by-step instructions provided in the

attached file, and add a response in each row of the table to indicate the changes that you have made. Please also address the additional marked-up edits we have proposed within the reporting summary. Ensuring that each point is addressed will help to ensure that your revised manuscript can be swiftly handed over to our production team.

We would hope to receive your revised paper, with all of the requested files and forms within two-three weeks. Please get in contact with us if you anticipate delays.

Nature Human Behaviour offers a Transparent Peer Review option for new original research manuscripts submitted after December 1st, 2019. As part of this initiative, we encourage our authors to support increased transparency into the peer review process by agreeing to have the reviewer comments, author rebuttal letters, and editorial decision letters published as a Supplementary item. When you submit your final files please clearly state in your cover letter whether or not you would like to participate in this initiative. Please note that failure to state your preference will result in delays in accepting your manuscript for publication.

In recognition of the time and expertise our reviewers provide to Nature Human Behaviour's editorial process, we would like to formally acknowledge their contribution to the external peer review of your manuscript entitled "Hippocampal neurons code individual episodic memories in humans". For those reviewers who give their assent, we will be publishing their names alongside the published article.

Cover suggestions

As you prepare your final files we encourage you to consider whether you have any images or illustrations that may be appropriate for use on the cover of Nature Human Behaviour.

ORCID

Non-corresponding authors do not have to link their ORCIDs but are encouraged to do so. Please note that it will not be possible to add/modify ORCIDs at proof. Thus, please let your co-authors know that if they wish to have their ORCID added to the paper they must follow the procedure described in the following link prior to acceptance:

Nature Human Behaviour has now transitioned to a unified Rights Collection system which will allow our Author Services team to quickly and easily collect the rights and permissions required to publish your work. Approximately 10 days after your paper is formally accepted, you will receive an email in providing you with a link to complete the grant of rights. If your paper is eligible for Open Access, our Author Services team will also be in touch regarding any additional information that may be required to arrange payment for your article. Please note that you will not receive your proofs until the publishing agreement has been received through our system.

Please note that *Nature Human Behaviour* is a Transformative Journal (TJ). Authors may publish their research with us through the traditional subscription access route or make their paper immediately open access through payment of an article-processing charge (APC). Authors will not be required to make a final decision about access to their article until it has been accepted. Find out more about Transformative Journals

[REDACTED]

Best regards,
Alex McKay
Editorial Assistant
Nature Human Behaviour

On behalf of

Jamie

Dr Jamie Horder
Senior Editor
Nature Human Behaviour

Reviewer #1:
Remarks to the Author:

These revisions satisfy all of my remaining concerns. Overall, the results seem pretty amazing to me, and they support the main claims made in this paper.

Reviewer #2:
Remarks to the Author:

The authors have sufficiently addressed all my remaining concerns.

Final Decision Letter:

Dear Dr Kolibius,

We are pleased to inform you that your Article "Hippocampal neurons code individual episodic memories in humans", has now been accepted for publication in *Nature Human Behaviour*.

Please note that *Nature Human Behaviour* is a Transformative Journal (TJ). Authors may publish their research with us through the traditional subscription access route or make their paper immediately open access through payment of an article-processing charge (APC). Authors will not be required to make a final decision about access to their article until it has been accepted. Find out more about Transformative Journals

An online order form for reprints of your paper is available at <http://www.nature.com/reprints/author-reprints.html>. All co-authors, authors' institutions and authors' funding agencies can order reprints using the form appropriate to their geographical region.

With best regards,

Jamie

Dr Jamie Horder
Senior Editor
Nature Human Behaviour